# Both fallopian tube and ovarian surface epithelium are cells-of-origin for high-grade serous ovarian carcinoma

Shuang Zhang[1]*, Igor Dolgalev [1], Tao Zhang [1], Hao Ran[1], Douglas A. Levine[1] & Benjamin G. Neel[1]*

The cell-of-origin of high grade serous ovarian carcinoma (HGSOC) remains controversial, with fallopian tube epithelium (FTE) and ovarian surface epithelium (OSE) both considered candidates. Here, by using genetically engineered mouse models and organoids, we assessed the tumor-forming properties of FTE and OSE harboring the same oncogenic abnormalities. Combined RB family inactivation and *Tp53* mutation in *Pax8* + FTE caused Serous Tubal Intraepithelial Carcinoma (STIC), which metastasized rapidly to the ovarian surface. These events were recapitulated by orthotopic injection of mutant FTE organoids. Engineering the same genetic lesions into *Lgr5* + OSE or OSE-derived organoids also caused metastatic HGSOC, although with longer latency and lower penetrance. FTE- and OSE-derived tumors had distinct transcriptomes, and comparative transcriptomics and genomics suggest that human HGSOC arises from both cell types. Finally, FTE- and OSE-derived organoids exhibited differential chemosensitivity. Our results comport with a dualistic origin for HGSOC and suggest that the cell-of-origin might influence therapeutic response.

[1] Laura and Isaac Perlmutter Cancer Center, NYU Langone Health, New York, NY 10016, USA. *email: zhangshuang0122@gmail.com; Benjamin.Neel@nyumc.org

High-grade serous ovarian cancer (HGSOC) is the most common and deadly ovarian epithelial cancer, causing ~70% of deaths[1]. HGSOC typically presents as a large ovarian mass accompanied by widespread peritoneal metastasis, but its cell-of-origin remains controversial[2–5]. Initially, HGSOC was thought to arise from invaginations of the ovarian surface epithelium (OSE) that result from normal ovulatory wounds. Trapped OSE within these so-called cortical inclusion cysts were believed to undergo Mullerian metaplasia and accumulate causal mutations[5–7]. Cortical inclusion cysts with columnar (Mullerian) epithelia and focal p53 immunoreactivity, but not frank carcinoma, have been reported, but such lesions are relatively rare[8,9]. Later, attention turned to the fallopian tube epithelium (FTE) as the likely cell-of-origin after serous tubular intra-epithelial carcinomas (STICs), defined as in situ neoplasms with increased proliferative capacity, TP53 mutation, and other characteristic markers, were reported in the fallopian tube fimbria of women with BRCA1/2 mutations undergoing risk-reducing salpingoophorectomy[10–12]. Subsequently, STICs were reported in up to 60% of sporadic HGSOC cases[13–15].

Multiple studies have attempted to define the cell-of-origin for HGSOC by genetic approaches. Essentially all HGSOCs have somatic mutations in TP53, and STICs, primary HGSOC, and metastases have the same TP53 mutation, implying a shared origin[16–19]. More comprehensive characterization (e.g., exome sequencing, copy number analysis, targeted amplicon sequencing, gene expression profiling) has shown that ovarian masses and distant metastases usually have the same truncal lesions as STICs, along with additional, often sub-clonal, abnormalities, suggesting that STICs are precursor lesions[13,20]. In addition, the transcriptomes of most HGSOCs more closely resemble normal FTE than OSE[21]. Nevertheless, up to 12% of HGSOCs show greater transcriptional similarity to OSE[21]; similar results were posted recently by others[22]. Proteomic analyses of ovarian cancer cell lines and primary tumors also suggest two subtypes of HGSOC, one FTE-derived and the other OSE-derived, with the latter having a worse prognosis[23]. A detailed re-analysis of public transcriptome data from multiple FTE, OSE, and HGSOC samples reached the same conclusion[24]. Moreover, some genomic studies have found that HGSOCs can metastasize to the FT and mimic STICs[25,26], consistent with a non-FTE origin, whereas metastases from other sites (e.g., uterine serous carcinoma) also can apparently mimic STICs[26–29].

Conclusions based on bioinformatic analysis are, by definition, inferences, and experimental evaluation of the relative roles of FTE and OSE in HGSOC pathogenesis remains lacking. Existing mouse models have failed to definitively resolve the cell-of-origin question. For example, injection of adenovirus expressing Cre recombinase (Ad-Cre) into the ovarian bursae of mice bearing conditional alleles of tumor suppressor genes and/or oncogenes, such as Myc;Tp53; Brca1[30], Pten;Pik3ca[31], Tp53;Rb1[32], or TgK18GT121 (N-terminal 121 amino acids of SV40 T antigen (T121) under the control of the keratin 18 promoter) and flTp53, +flBrca1[33], result in HGSOC-like tumors. These studies assume that bursal injection specifically targets the OSE, but others have argued that such adjacent tissue is also infected, including the FT or even the uterus[34]. Mice with Pten deletion, Tp53 deletion or mutation, and Brca1/2 deletion driven by the Pax8 promoter, which is expressed selectively in FTE secretory cells (not in OSE), develop STIC-like lesions and eventually, widely metastatic HGSOC[35]. Similar results were seen with in mice expressing SV40 large T-antigen (TAg)[36], or with simultaneous mutation of Brca1, Tp53, Rb1, and Nf1, in Ovgp1-expressing secretory FTE[37]. These studies establish that FTE can be the cell-of-origin for HGSOC, but whether the same (or other) oncogenic events can cause HGSOC to arise in OSE was not addressed.

Different cells-of-origin could, along with distinct genomic abnormalities, contribute to inter-tumor heterogeneity and clinical behavior[38–40]. Here, we use lineage-specific Cre recombinase (Cre) lines and FTE-derived and OSE-derived organoids to ask whether introducing the same genetic defects into FTE or OSE can cause HGSOC, and if so, how the cell-of-origin affects tumor biology.

## Results

**Perturbing Tp53 and the RB family in Pax8+ cells cause STIC and metastasis.** TgK18GT121 (hereafter, T121) mice harbor a bacterial artificial chromosome (BAC) containing a loxP-GFP-stop-loxP (LSL) T121 cassette inserted into the mouse cytokeratin (CK) 18 gene. This construct ensures that T121 (the N-terminal 121 amino acids of SV40 T antigen, which inactivates all RB family members) is only expressed upon exposure to Cre. Ad-Cre injection into the ovarian bursa of mice harboring TgK18GT121 and Tp53 deletion or heterozygotic Tp53R172H expression causes HGSOC[33]. These findings were interpreted as showing that OSE gives rise to HGSOC, but the mouse FT (oviduct) is also housed within the bursa and could have been infected in these experiments. Indeed, Ad-Cre injection into mice bearing a Cre-inducible lacZ allele (Rosa26-lacZ) resulted in mosaic staining for the lacZ gene product β-galactosidase (β-gal) in FTE, in addition to the expected strong β-gal staining in OSE. Moreover, Ad-Cre injection into the bursae of T121;Tp53R172H mice evoked excess proliferation of both FTE and OSE, leaving the cell-of-origin in this model unclear (Supplementary Fig. 1).

To resolve this ambiguity, we generated T121;Tp53R172H mice that selectively express Cre in OSE or FTE. Pax8rtTA mice reportedly enable doxycycline (Dox)-inducible gene expression in secretory cells of the FTE[35]. To confirm the lineage fidelity of this line, we studied Pax8rtTA;TetOCre:Rosa26-tdTomato and Pax8rtTA;TetOCre:Rosa26-LacZ reporter mice (Fig. 1a, Supplementary Fig. 2). Following 2 weeks of Dox administration (0.2 mg/ml in drinking water ad libitum), histologic analysis of Pax8rtTA;TetOCre:Rosa26-Lacz mice revealed strong β-gal reactivity in FTE, but no staining in the ovary, including the OSE (Fig. 1b). After Pax8rtTA;TetOcre;Rosa26-tdTomato mice were exposed to Dox for 2 days (pulse), Tomato+ cells were found to co-localize with PAX8+ secretory FTE cells (Supplementary Fig. 2a, b). When similarly treated mice were switched to Dox-free water for 2 months (chase), both secretory (PAX8+) and ciliated (acetylated α-tubulin+) cells were Tomato+ (Supplementary Fig. 2c, d). These data confirm that Pax8rtTA drives expression in FTE, but not OSE, and that PAX8+ FTE cells give rise to ciliated cells in vivo, consistent with a recent study[41].

We next generated Pax8rtTA;TetOcre;T121 (PTT) mice, which have Dox-inducible expression of T121 alone, Pax8rtTA;TetOcre; Tp53R172H/fl (PTP) mice, which allow Dox-inducible generation of Tp53R172H hemizygosity, and Pax8rtTA;TetOcre;Tp53R172H/fl; T121 (PTPT) mice, which show Dox-dependent combined T121 expression/Tp53R172H hemizygosity, all specifically in Pax8+ cells (Fig. 1c). A GFP reporter is embedded within the LSL cassette of the T121 transgene (Fig. 1c), allowing assessment of deletion efficiency. As expected, in control PTPT mice, GFP was detected throughout the FTE, which also showed little proliferation (Ki67 staining). After 2 weeks of Dox administration, Cre activation in Pax8+ cells had resulted in deletion of the LSL cassette, indicated by the appearance of GFP− cells. The newly generated GFP− (T121-expressing) cells, but not GFP+ (Pax8−) cells, were Ki67-positive, demonstrating that T121 evokes FTE proliferation (Fig. 1d). At 1 month after Dox administration was completed, FTE from all (10/10) treated PTPT mice showed morphological features of STIC, including secretory cell

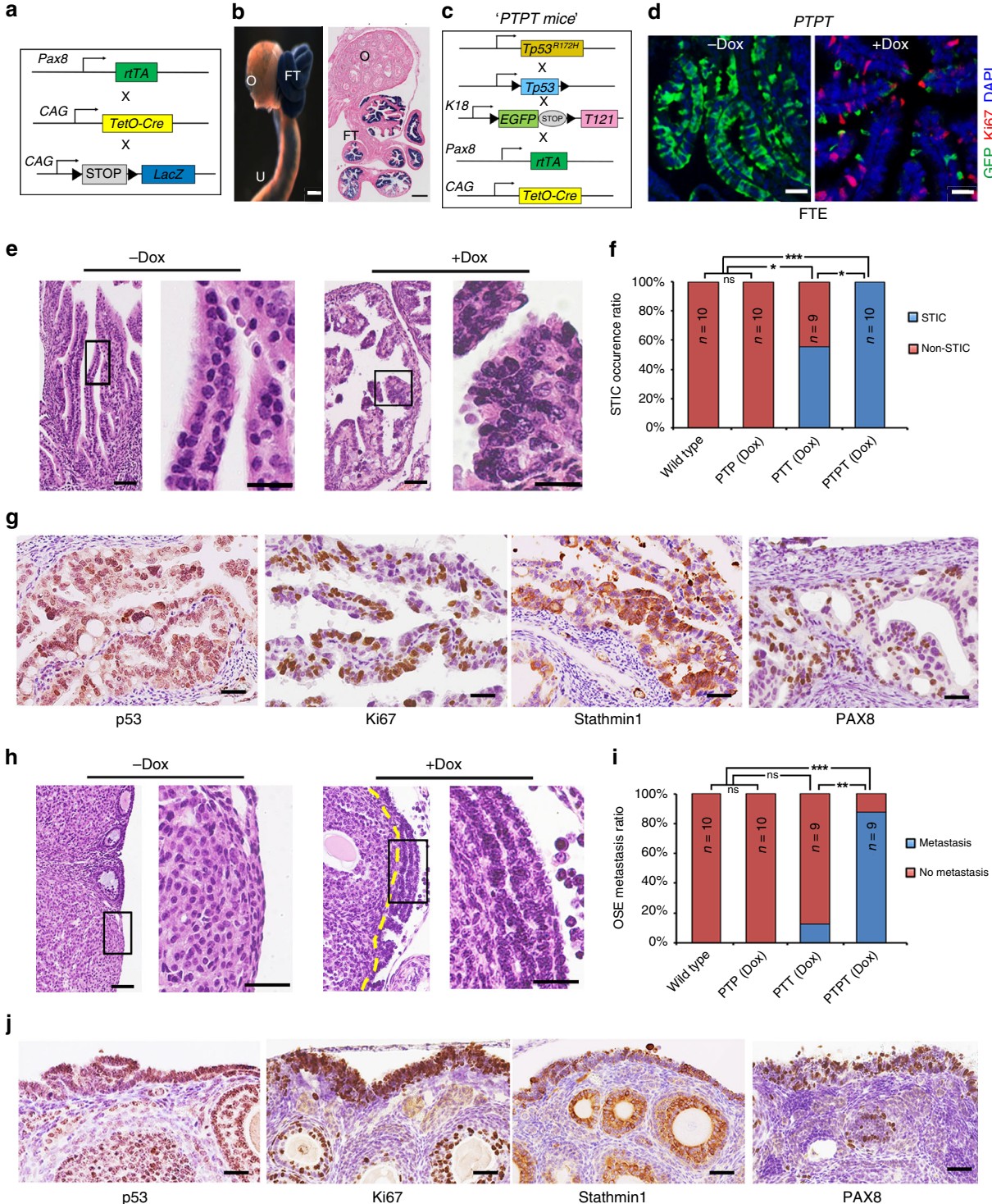

**Fig. 1** Combined *Tp53* mutation/RB family inactivation in fallopian tube epithelium causes HGSOC. **a** Schematic of *Pax8rtTA*, *TetOCre*, and *Rosa26-lacZ* alleles. **b** Whole mount (left panel) and 10×-magnified histological section (right panel) of genital tracts of *Pax8rtTA;TetOCre;Rosa26-LacZ* female mice, treated with Dox (2 mg/ml in drinking water for 2 weeks), followed by X-gal staining and eosin counterstaining. Scale bar: 100 μm **c** Schematic of *Pax8rtTA; TetOcre;Tp53$^{R172H/-}$;T121* (PTPT) mice. **d** Immunofluorescence staining for GFP (marker for keratin 18+ FTE cells, green), Ki67 (proliferation marker, red) and DAPI (nuclear stain, blue), with or without Dox treatment, as in **b**. Scale bar: 10 μm. **e** Representative H&E-stained section of fallopian tube from PTPT mice collected 1 month after completion of Dox treatment (2 mg/ml in drinking water for 2 weeks). Boxed regions in left panels are magnified in the right panel of each pair. Scale bar: 50 μm. **f** % mice of each genotype with STIC, assessed 1 month after completion of Dox treatment, as above. **g** IHC for STIC markers in representative FTE sections of FTE from PTPL mice. Scale bar:10 μm. **h** Representative H&E-stained section from PTPT ovaries, assessed 1 month after completion of Dox treatment as above or in mice left untreated for the same time interval. Boxed regions in left panels are magnified in the right panel of each pair. Yellow line indicates the border between OSE and underlying cells. Scale bar: 50 μm. **i** % of mice of each genotype with ovarian metastasis (assessed as in **h**). **j** IHC for HGSOC markers in representative ovary sections from PTPL mice. Scale bar: 10 μm. ns, not significant. *$P < 0.05$, **$P < 0.01$,***$P < 0.001$, Fischer's exact test.

proliferation, loss of polarity, cellular atypia, serous histology, and expression of characteristic immunohistochemical markers (Fig. 1e–g), as well as metastasis to the ovarian surface (Fig. 1h; compare boxed regions). We did not detect STIC in PTP mice (0/10), although 5/9 PTT mice showed FTE transformation, suggesting that RB family inactivation alone can cause STIC. Nevertheless, combined *Tp53* perturbation/T121 expression significantly increased FTE transformation (from 55% to 100%) and metastasis (from 0% to 88%) to the ovary (Fig. 1i, j). Immunohistochemistry (IHC) showed HGSOC-related features, including TP53 accumulation, proliferation, and PAX8 expression in primary lesions (Fig. 1g) and ovarian metastases in PTPT mice (Fig. 1j).

Unexpectedly, nearly all PTT and PTPT mice died within 2 months (Supplementary Fig. 3a), but their demise was not due to peritoneal masses/obstruction, as expected from lethal HGSOC. Instead, their thymi were massively enlarged, owing to increased numbers of thymic epithelial cells (Supplementary Fig. 3b and c), which led to breathing difficulty and ultimately, respiratory arrest. Combined *Rb/p130* deletion/*p107* heterozygosity causes a similar phenotype[42], and surprisingly, re-analysis of *Pax8rtTA;TetOCre;Rosa26-tdTomato* mice showed that *TetOCre* is active in thymic epithelial cells (although not in FTE) even without Dox (Supplementary Fig. 3d). The unanticipated leakiness of *TetOCre* in the thymic epithelium precluded us from asking if *Pax8*-driven T121 expression/*Tp53* perturbation leads to widespread metastasis, although the early ovarian studding and our organoid studies (see below) make this highly likely.

**FTE organoids recapitulate fallopian tube differentiation and transformation in vitro.** Organoids initiated from adult stem cells can be propagated long-term and are useful for cancer modeling[43,44]. Culture conditions for human FTE organoids have been established[45], but OSE organoids have not been reported, nor have human or mouse organoids been used to model HGSOC. We developed serum-free, defined media that allow mouse FTE and OSE (see below) organoids to be passaged indefinitely (>30 passages) in Matrigel-based media, cryopreserved, and re-established in culture (see the "Methods" section). By 48 hr after seeding FACS-purified EPCAM−/CD45− cells, small (1–5 μm in diameter), round, cystic spheres appeared (EPCAM+CD45−), and they expanded rapidly to form large (100–1000 μm in diameter) hollow cysts. After 10 days, epithelial invagination resulted in mucosal folds, morphologically recapitulating FT architecture (Fig. 2a). Like their tissue of origin, organoids cultured > 7 days contained secretory and ciliated cells (Fig. 2b).

To ask if this system could model HGSOC, we established organoid cultures from FTE of wild type, PTP, PTT, and PTPT mice. Dox-treated PTT and PTPT organoids induced *Cre*, as indicated by loss of GFP after three passages (Fig. 2c). Dox-treated PTPT organoids also exhibited frank dysplasia, TP53 accumulation, and increased staining for the DNA damage marker γH2AX (Fig. 2d, Supplementary Fig. 4a). Consistent with our GEMM results (Fig. 1), PTT and PTPT organoids proliferated more rapidly and were larger than controls (Fig. 2e, f, Supplementary Fig. 4b, c). By contrast, organoid-forming efficiency was not increased by *Tp53* mutation and/or RB family inactivation, indicating that these genetic abnormalities do not enhance FTE self-renewal in vitro (Supplementary Fig. 4d). STIC and HGSOC typically are PAX8+ (although staining can be mosaic), and they do not express ciliated cell markers[46]. Similarly, ciliated cells were undetectable in PTT or PTPT organoids, suggesting that the RB family inactivation restricts FTE differentiation to the secretory lineage (Fig. 2g, h).

*Tp53^{R172H/−}* mutation did not impair differentiation, although it enhanced invasive capability (Fig. 2i, j). Thus, by using FTE organoids, various aspects of the malignant phenotype can be attributed to specific genetic defects.

**Mutant FTE organoid transplantation can recapitulate HGSOC progression.** We reported previously that human HGSOC cells implanted into the mouse mammary fat pad (MFP) recapitulate HGSOC histomorphology, inter-tumor and intra-tumor heterogeneity, and patient drug response[47,48]. As an initial test of their tumor-forming capacity, we injected wild type (WT) and mutant FTE organoids into the MFPs of *nu/nu* mice (Fig. 3a). WT and *Tp53^{R172H/−}* organoids formed glandular structures comprising simple cuboidal cells that persisted for at least 4 weeks, but these disappeared (*n* = 0/9 for wild type or PTP organoids) by 2 months (Fig. 3b–d). By contrast, PTPT organoids formed palpable (>2 mm) masses of high grade, poorly differentiated adenocarcinoma with prominent cellular/nuclear pleomorphism, and frequent mitoses (Fig. 3d). PTT organoids also gave rise to HGSOC (3/8 mice), but PTPT organoids were more tumorigenic (9/10 mice, Fig. 3b), consistent with the GEMM results (Fig. 1f, i). The ovary is thought to provide trophic signals that enable HGSOC growth and metastasis[49], so we also injected organoids into the ovarian fat pad. PTT and PTPT organoids both gave rise to primary tumors, but when assessed at 3 months, 7/10 mice injected with PTPT organoids had developed ascites, sometimes hemorrhagic, with widespread peritoneal studding, similar to the human disease. By contrast, PTT-derived organoids (0/10 lines) did not show evident metastasis at 9 months post-injection (Fig. 3e, f). Metastases expressed HGSOC markers, including PAX8, p53, Ki67, γH2AX, Stathmin1, and p16 (Fig. 3g). Hence, FTE organoids recapitulate HGSOC biology morphologically and molecularly.

**Combined *Tp53* mutation and RB family inactivation in *Lgr5+* OSE cells also causes HGSOC.** As an initial test of whether OSE also can be an HGSOC cell-of-origin, we injected Ad-Cre into the ovarian bursae of *Tp53^{R172H/fl};T121* mice, and performed salpingectomies 3 days later (Supplementary Fig. 5a). Gross inspection and histology confirmed that the FT had been removed (Supplementary Fig. 5b). Nevertheless, 3 months later, the OSE of similarly treated mice showed abnormal morphology and hyperproliferation (Supplementary Fig. 5c), prompting us to more carefully evaluate oncogenesis in OSE.

*Lgr5+* embryonic and neonatal populations establish the OSE lineage and the fimbrial FTE[50]. In adult mice, however, *Lgr5* expression is concentrated in the ovarian hilum, together with the stem cell markers CD133 and ALDH, and lineage-tracing experiments using *Lgr5Cre^{ERT2}* mice show that *Lgr5+* cells can repopulate the entire OSE[51]. To confirm the lineage fidelity of this *Cre* line and assess its expression in adult FTE, we generated *Lgr5Cre^{ERT2};Rosa26-tdTomato* mice. On Day 2 after administration of a single dose of 4-OHT (to induce Cre activity), only a few Tomato+ cells were found in OSE. By 4 months, though, these cells populated a significant fraction of OSE. Importantly, no FTE expression was observed (Fig. 4a, b).

We next generated and compared *Lgr5-Cre;Tp53^{R172H/fl};T121* (LPT), *Lgr5-Cre;Tp53^{R172H/fl}* (LP), and *Lgr5-Cre;T121* (LT) mice (Fig. 4c). All mice received a single intraperitoneal dose of 4-OHT. As expected, LP mice (*n* = 0/10) did not develop tumors, and only 1/12 LT mice showed an obvious ovarian mass over the next 18 months. By contrast, all (11/11) LPT mice developed large, palpable, abdominal masses by 11 months post-4-OHT injection (Fig. 4d, e). Autopsies revealed markedly enlarged, hemorrhagic ovaries, with 16% (*n* = 2/11) showing multifocal

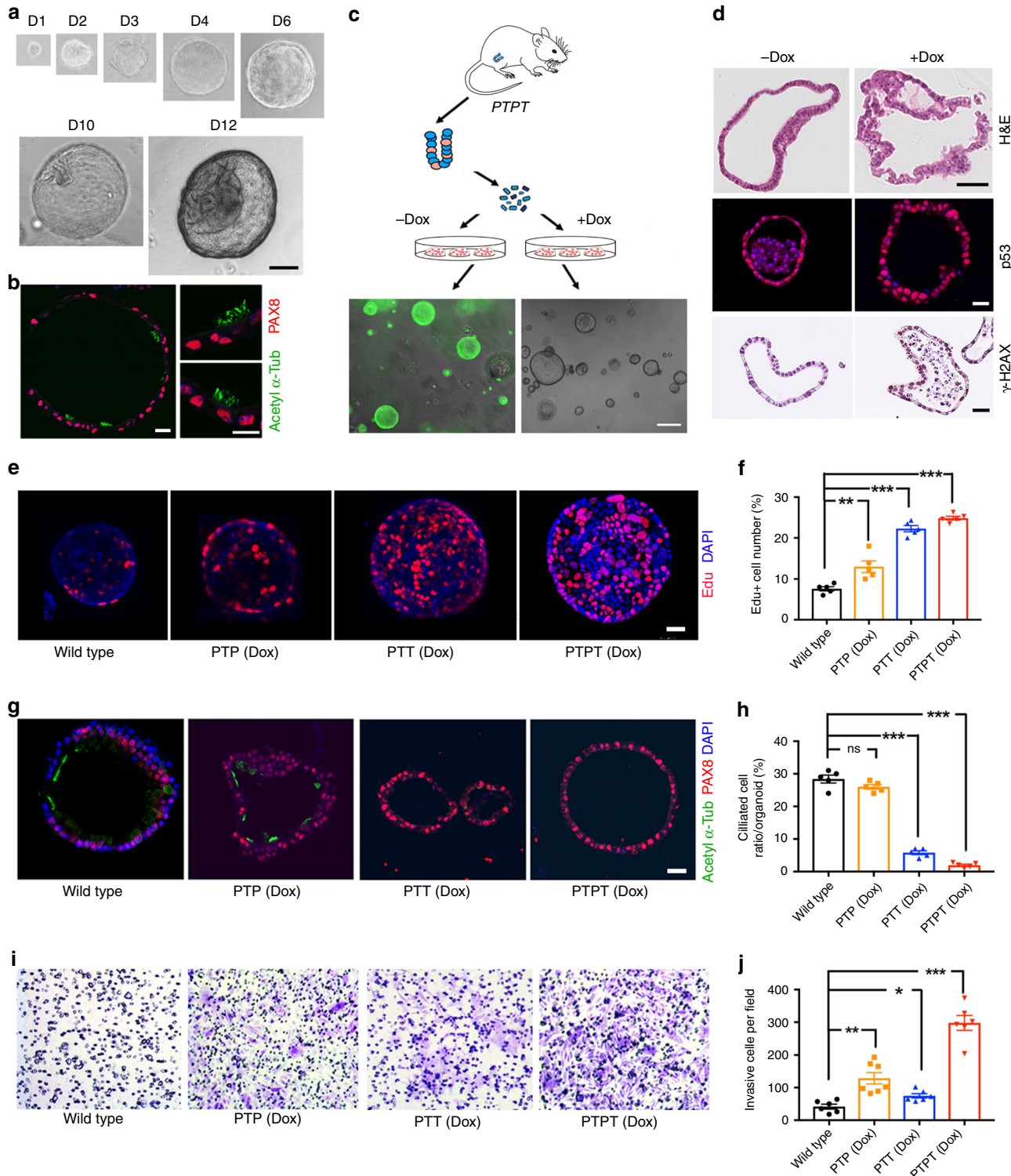

peritoneal carcinomatosis and ascites (Fig. 4d, f). Histology and IHC revealed poorly differentiated adenocarcinoma with papillary and micropapillary/filigree morphology (Fig. 4g), resembling human HGSOC. Notably, these tumors showed little PAX8 staining.

As an additional test of whether *Lgr5+* OSE cells directly evoke HGSOC transformation, we generated compound LPT;*Rosa26-tdTomato* mice (Fig. 5a), administered one dose of 4-OHT, and analyzed mice at 3, 6, or 9 months (Fig. 5b). At 3 months after Cre activation, neoplasia had developed from *Lgr5+* cells (Fig. 5c).

These lesions expanded, and at 6 months, >50% of the ovarian surface was covered by malignant Tomato+ cells; at 9 months, the surface was almost entirely overrun (Fig. 5c–f). Histology again showed multi-villus neoplasia (Fig. 5d, g), and IHC revealed highly proliferative tumors expressing HGSOC markers, including Wilms' tumor 1 (WT1) and Stathmin1 (Fig. 5g). Neither hyperproliferation nor neoplasia was evident in the FT of LT mice (Supplementary Fig. 6a), and the OSE-derived, LPT tumors also were PAX8-negative (Supplementary Fig. 6b). As expected[52], we also saw Tomato+ clones in the intestines of *Lgr5Cre*$^{ERT2}$;

**Fig. 2** Establishment and characterization of FTE organoids **a** Bright field images of organoid developing from a single FTE cell after the indicated days of culture. Scale bar, 20 μm. **b** Immunofluorescence staining for ciliated cell marker acetyl-α-tubulin (green) and secretory cell marker PAX8 (red) in FTE organoid after 7 days in culture. Scale bar, 2 μm. **c** Top: Schematic showing generation of FTE organoids from PTPT mice. Bottom: GFP in organoids with or without Dox treatment; Dox (500 ng/ml) was added after seeding, and GFP− cells were removed by FACS 4 days after Dox addition. The organoid in the right panel was photographed before Dox addition; the organoids in the left panel were photographed after Dox (see the "Methods" section for details). Scale bar, 10 μm. **d** Sections of PTPT organoids after two consecutive passages, with or without Dox treatment (500 ng/ml), stained with H&E or subjected to immunofluorescence or IHC staining for the indicated markers. Scale bar, 20 μm. **e** Organoids established from the indicated mice, incubated with EdU (2 μM) and DAPI (1 μg/ml) for 2 h on day 7 of culture and visualized by immunofluorescence; scale bar, 20 μm. **f** % EdU-positive cells in organoids established from the indicated mice. **g** Representative immunofluorescence images of PAX8 (red) and acetylated-α-tubulin (green) in the indicated organoids at day 7 of culture; scale bar, 20 μm. **h** % ciliated cells (aceylated-α-tubulin+) in organoids established from the indicated mice. **i** Representative images of the bottom surface of Transwell units seeded with the indicated organoids. **j** Quantification of invasive cells in **i**. Data represent mean ± SEM from three mice of each genetic background. *$P < 0.05$, **$P < 0.01$, ***$P < 0.001$, Tukey's multiple comparison test. Source data are provided as a Source Data file.

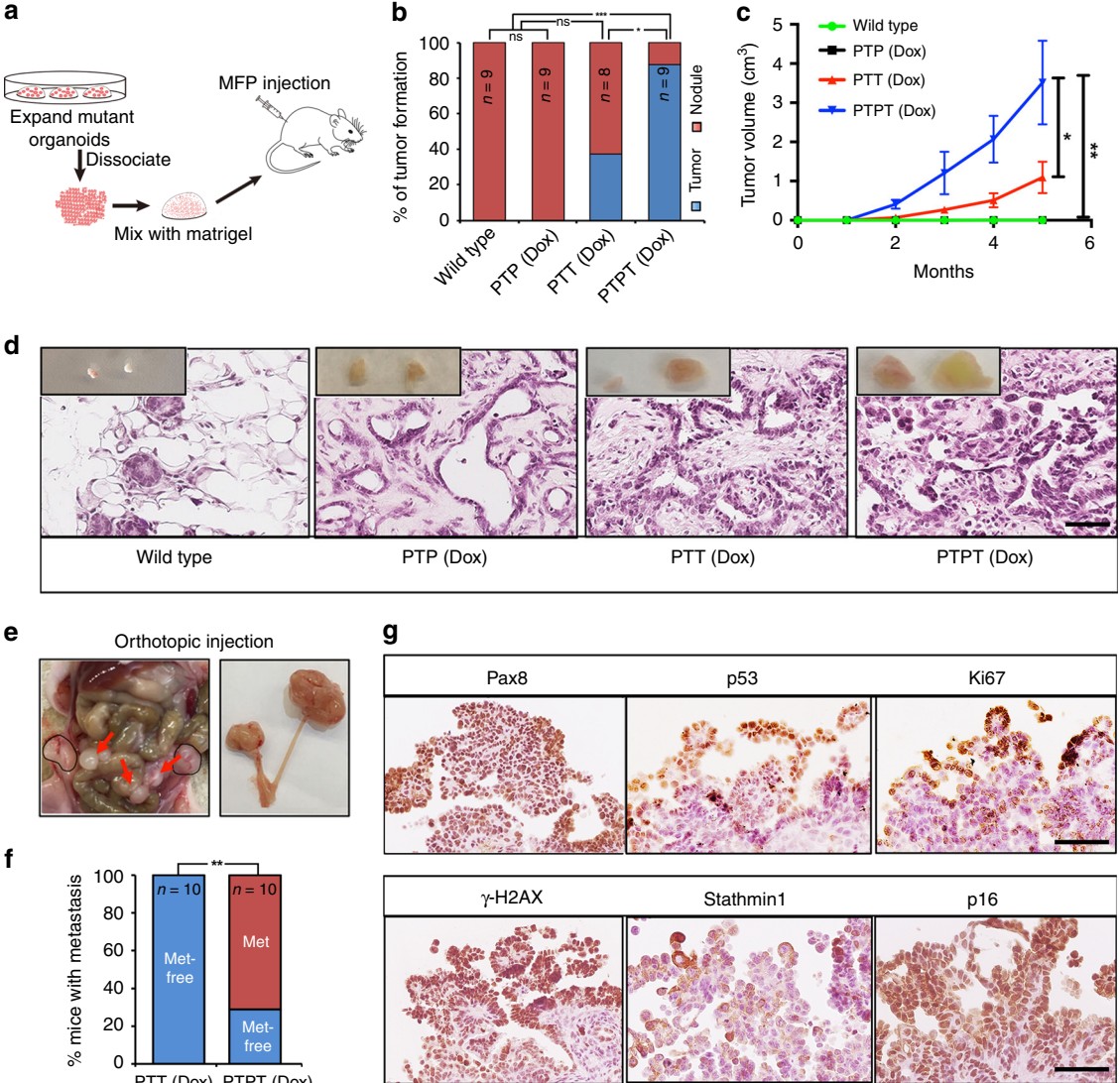

**Fig. 3** Transplanted organoids recapitulate HGSOC progression and metastasis. **a** Scheme depicting mammary fat pad (MFP) transplantation. **b** % of tumors formed in MFPs within 6 months after injection of $10^5$ cells from wild type, PTP, PTT, and PTPT organoids (all Dox-treated); ns, not significant, *$P < 0.05$, ***$P < 0.001$, Fischer's exact test. **c** Graph showing the average tumor volume in MFPs injected with organoids of the genotypes indicated in **b**. Data represent mean ± SE, *$P < 0.05$, **$P < 0.01$, two-way ANOVA; Source data are provided as a Source Data file. **d** H&E-stained sections of MFPs from mice injected with $10^5$ cells from the indicated organoids, 3 months after injection; inserts show gross appearance of nodules/tumors. Scale bar, 50 μm. **e** Left panel: exposed abdomen of mouse 3 months after orthotopic injection with $10^5$ PTPT organoid cells, showing ovarian masses (circles) and widespread peritoneal metastasis (arrows). Right panel: genital duct dissected from the left panel shows large tumors on ovary. **f** % of mice injected with Dox-treated PTT (0%, $n = 10$) and PTPT (30%, $n = 10$) organoids that develop peritoneal metastasis, assessed at 6 months following injection of $10^5$ organoid cells; **$P < 0.01$, Fischer's exact test. **g** H&E-stained sections and IHC for the indicated HGSOC markers in omental metastases; scale bar, 50 μm.

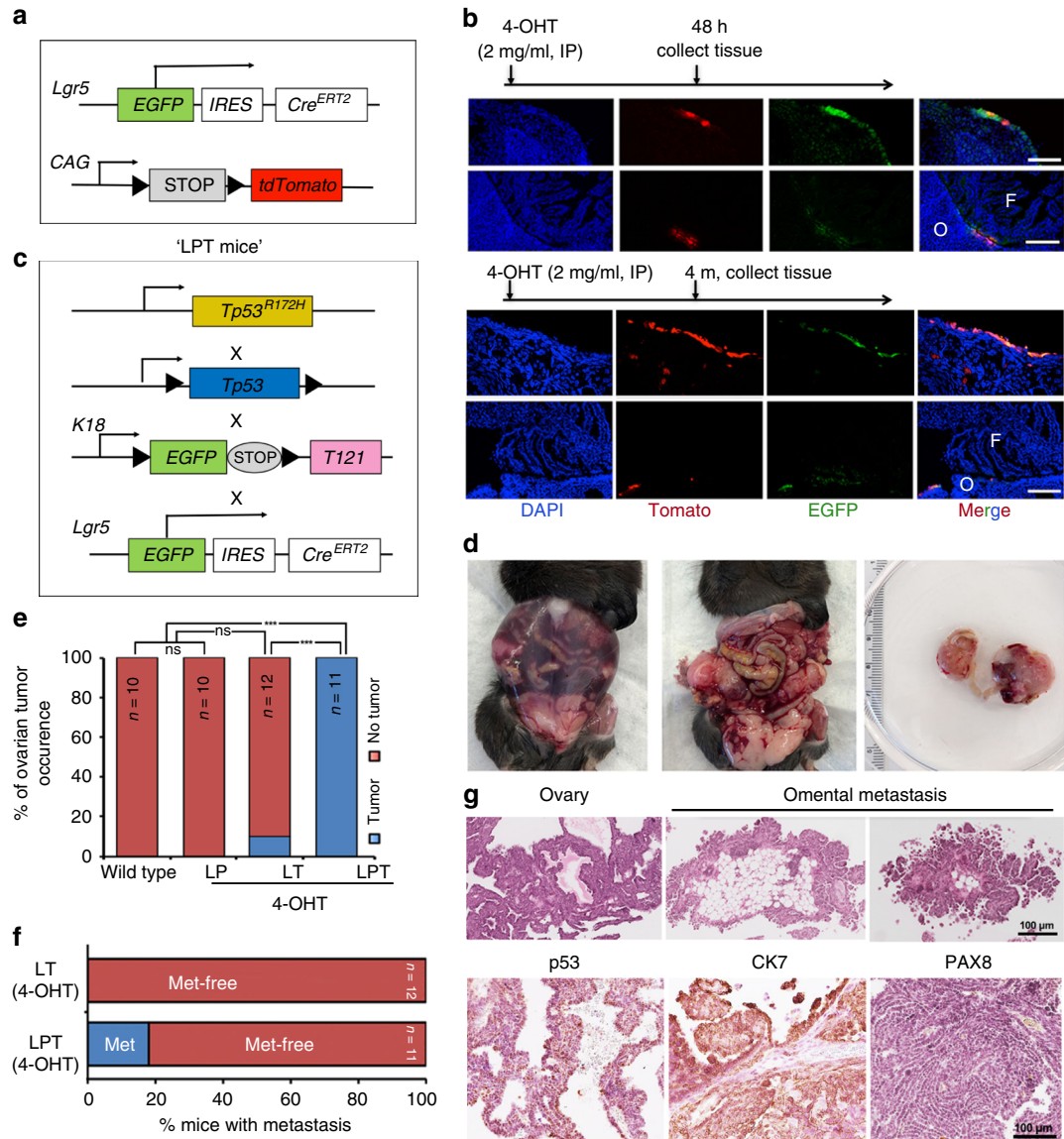

**Fig. 4** Combined *Tp53* mutation/RB family inactivation in *Lgr5+* OSE also causes HGSOC. **a** Scheme depicting *Lgr5Cre;Rosa26-tdTomato* mice. **b** EGFP co-immunostaining of Tomato+ OSE clone in ovary and fallopian tube sections, 48 h or 4 months post 4-OHT induction of *Lgr5-Cre;Rosa26-tdTomato* mice; scale bars, 50 μm; O ovary; F fallopian tube. **c** Scheme depicting *Lgr5Cre^{ERT2};T121;Tp53^{R172H/−}* (LPT) mice. **d** Exposed abdominal cavity of LPT mouse at 11 months after 4-OHT treatment, showing abdominal distention from ascites, large ovarian tumors and peritoneal studding. **e** % of mice of the indicated genotypes with ovarian tumors at 18 months post 4-OHT induction, ***$P < 0.001$, Fischer's exact test. **f** % LT and LPT mice with peritoneal metastasis at 18 months after 4-OHT treatment. **g** Representative p53, CK7, and PAX8 staining (by IHC) in metastatic tumor from LPT mouse.

*Rosa26-tdTomato* mice after 2 days of 4-OHT, and these expanded to populate the intestinal villi at 4 months post-4-OHT administration (Supplementary Fig. 6c). However, intestinal carcinoma was not observed in 4-OHT-treated LPT mice at 11 months (Supplementary Fig. 6d).

**OSE-derived organoids support an ovary origin for ovarian cancer.** We also developed conditions to culture organoids from single OSE cells (Fig. 6a). OSE organoids expressed epithelial markers, such as E-cadherin, but unlike their FTE counterparts, they did not express PAX8 (Fig. 6b). As noted above, *Lgr5+* cells reportedly have stem/progenitor activity; indeed, GFP^{hi} (*Lgr5*-expressing) cells accounted for almost all organoid-forming capability in OSE ($P < 0.001$, Fig. 6c). We next generated organoids from wild type, LP, LT, and LPT OSE, and added 4-OHT to the media for 48 h to induce T121 expression in LT and LPT

organoids (Fig. 6d). Organoid diameter was not affected, but mutant organoids were denser and contained more cells (Fig. 6e). Although organoid-forming capacity was similar in the four groups, LPT ($P < 0.001$), and to a lesser extent, LT ($P < 0.01$) organoids, grew faster than the others (Fig. 6g). Consistent with their cognate GEMMs (Fig. 4f), only LPT organoids induced tumors upon MFP (Fig. 6h) or orthotopic (Fig. 6h right panel and Fig. 6i) injection. Importantly, organoids and the tumors that they evoked displayed similar histology and marker expression (Fig. 6j, k).

To compare the tumorigenicity of transformed FTE and OSE, we injected equal numbers of induced (GFP−) cells from PTPT (FTE, Dox-induced) and PTL (OSE, 4-OHT induced) organoids into the ovarian fat pads of *nu/nu* mice. Latency and survival were longer for OSE-derived tumors (Supplementary Fig. 7a). We also infected organoids from *Tp53^{R172H/fl};T121* mice with Ad-Cre, purified GFP− (i.e., compound mutant) cells 6 days later,

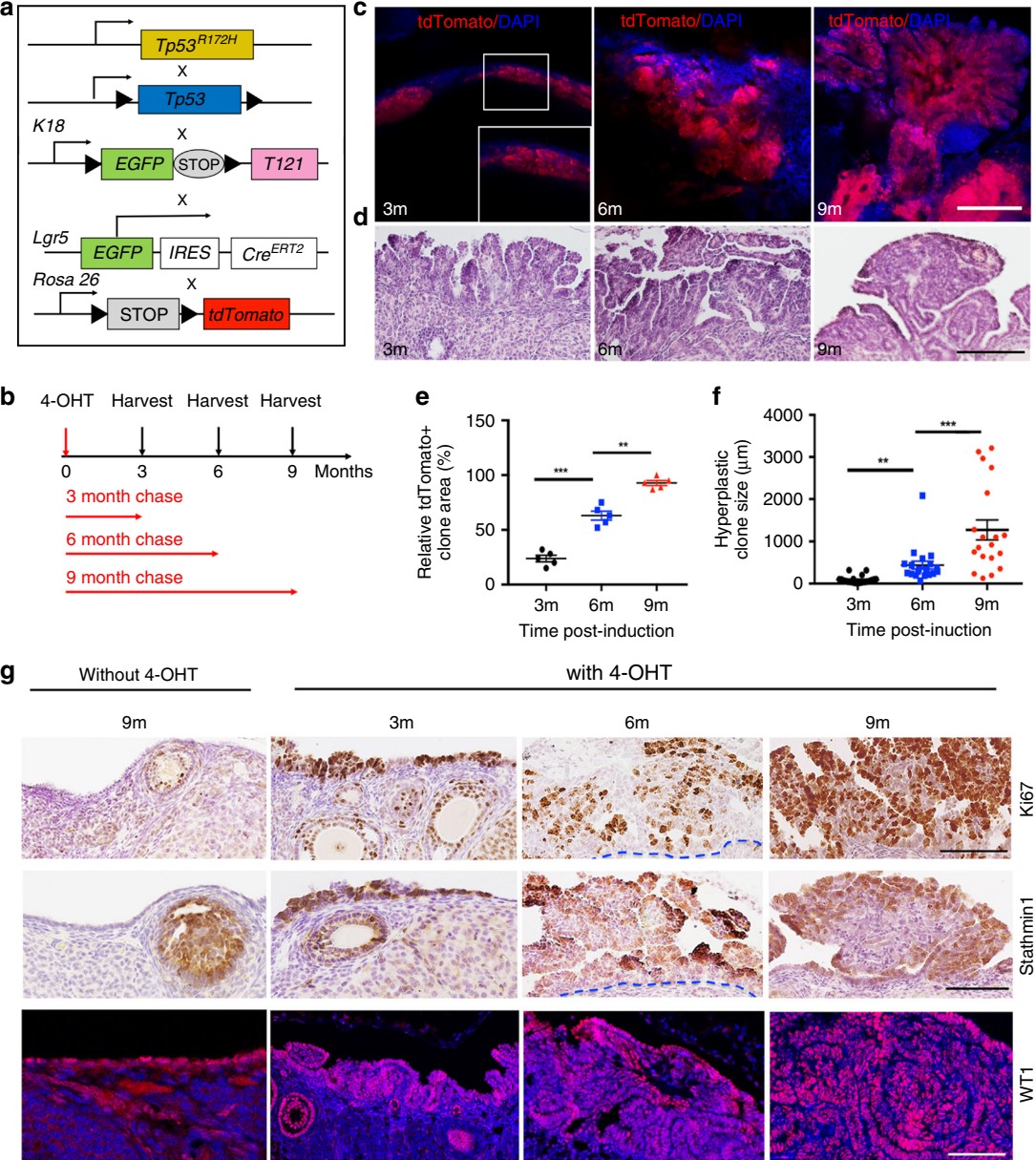

**Fig. 5** *Lgr5+* cells can initiate tumors with markers of HGSOC. **a** Schematic depicting crosses of *Tp53^R172H/−^;T121;Lgr5Cre;* and *Rosa26-tdTomato* mice. **b** Lineage tracing scheme for *Lgr5Cre;Rosa26-Tdtomato* mice. **c** Whole mounts of ovaries from *LPT;Rosa26-Tdtomato* mice induced with 4-OHT at 6 weeks (pulse), followed by sacrifice and analysis at the indicated times (chase), showing Tomato (red), and DAPI (blue) fluorescence; region within the white box in left top panel is magnified below. Note expansion of isolated Tomato+ cells into Tomato+ clones. Scale bar, 50 μm. **d** Corresponding H&E-stained sections from ovaries in **c**; **e** Relative areas of Tomato+ OSE clones at the indicated times after 4-OHT induction (chase times); clone size was determined by measuring the longest length of a discrete Tomato+ clone. Each circle represents a distinct Tomato+ clone, with n = 3 ovaries analyzed/time point, and a total of 180 clones counted (full data are provided as a Source Data file). **f** Quantification of hyperplastic clone size on the ovarian surface, determined by measuring the longest 'length' of epithelial protrusions (tumor cells) from the border with stromal cells. For each time point, 20 cross-sections from three ovaries were counted. Data represent mean ± SEM; *P < 0.05, **P < 0.01, and ***P < 0.001, Tukey's multiple comparison test; Source data are provided as a Source Data file. **g** Representative IHC (Ki67 and Stathmin-1) and immunofluorescence (WT-1) staining for HGSOC markers in OSE from LPT mice at the indicated times after 4-OHT induction or without treatment; scale bars: 100 μm.

and injected $10^5$ each into MFPs (Supplementary Fig. 7b). Again, OSE-derived tumors had a slower growth rate and longer latency ($P < 0.05$ at 4 months, $P < 0.001$ at 5 months, two-way ANOVA, Supplementary Fig. 7c) than their FTE counterparts.

**Similarities and differences between transcriptomes and genomes of FTE-derived and OSE-derived HGSOC.** We used RNA sequencing (RNAseq) to explore the molecular basis for the distinct behavior of FTE-derived and OSE-derived HGSOC. Five

independent tumors from FTE-derived PTPT organoids (T-FT) were compared to three tumors from OSE-derived organoids (T-O), all at 6 months post-injection. Pooled normal OSE (N-O) and FTE (N-FT) samples (three each) served as controls. FTE (tumor and normal) and OSE (tumor and normal) samples segregated by unsupervised clustering (Fig. 7a). Of the 8623 differentially expressed genes (DEGs) in FTE- versus OSE-derived tumors ($P_{adj} < 0.05$), 3641(42%) were differentially expressed in cognate normal tissue (intersection between N-O vs. N-FT, purple ovals,

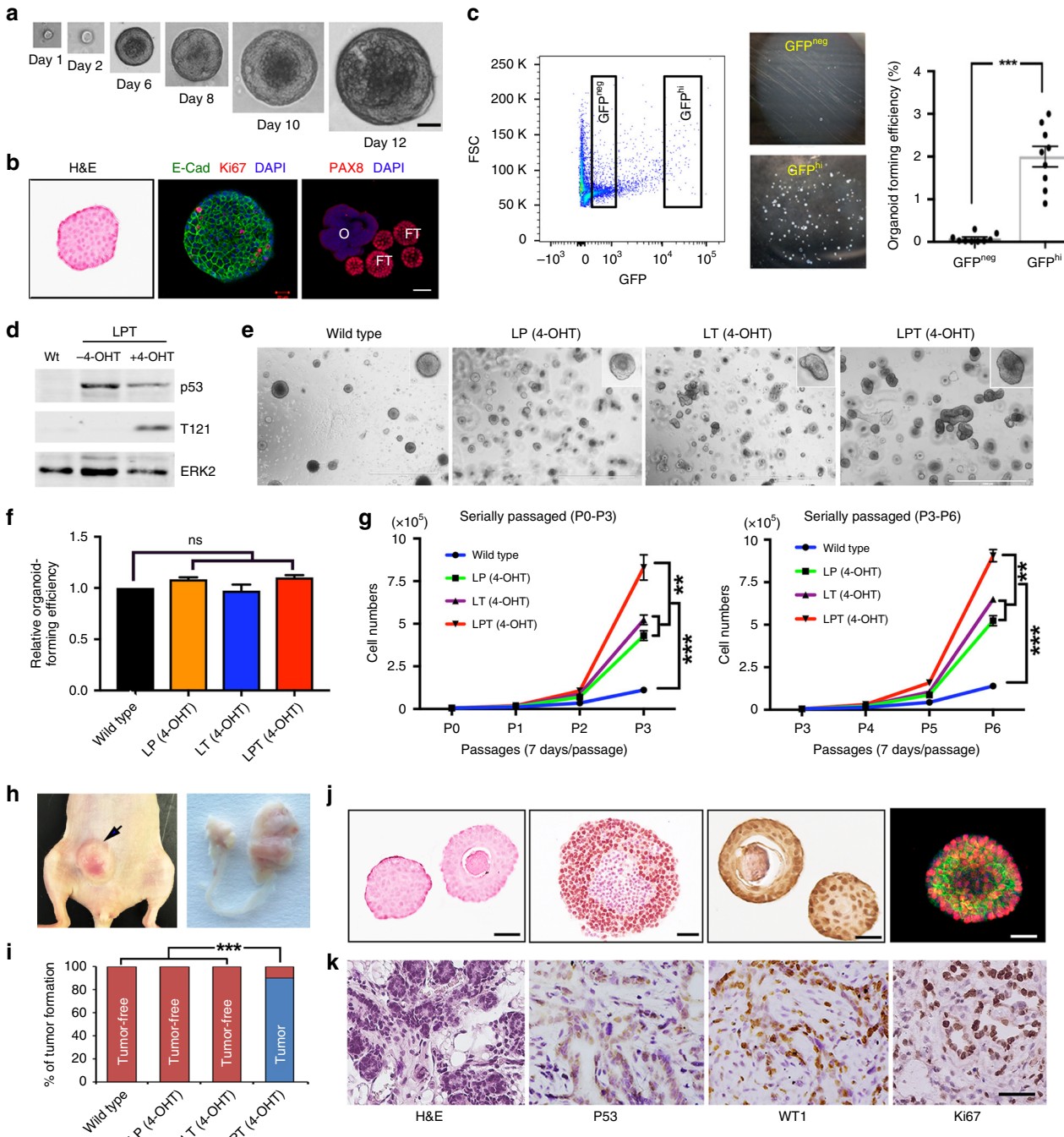

**Fig. 6** Transplanted OSE organoids also can give rise to HGSOC. **a** Representative serial images of OSE organoid at the indicated times after seeding; magnification: ×10, scale bar, 20 μm. **b** Left panel: H&E stain of OSE organoid; Middle panel: IF for E-cadherin and Ki67; Right panel: IF for PAX8 in FTE (FT) and OSE (O) organoids. Note that only FTE organoids are strongly PAX8+. Scale bar, 20 μm. **c** Representative flow cytometric plot, showing gates used to purify EGFP[hi] and EGFP[neg] populations from *Lgr5-GFP* OSE cells (Left panel). Bright field pictures show organoids that developed from FACS-purified cells (Middle panel). Right panel shows organoid-forming efficiency of the two populations. Data indicate means ± SEM, ***p < 0.001, unpaired *t* test. **d** Representative immunoblot for T121 in LPT organoids without 4-OHT treatment and 2 weeks after 4-OHT induction; ERK2 serves as a loading control. Wt: wild type organoid. **e** Micrographs of typical OSE organoids from mice of the indicated genotypes after 6 days culture. **f** Relative organoid-forming efficiency of OSE cells from mice of the indicated genotypes; Data represent means ± SEM; ns not significant, Tukey's multiple comparison test. **g** Growth curves of OSE organoids from the indicated mice; Cells per well were counted at each passage from P1–P3 and P4–P6. Organoids from three independent mice for each group were used, each in duplicate. (Source data are provided as a Source Data file.) Data represent means ± SEM, **P < 0.01, ***P < 0.001, two-way ANOVA. **h** Gross pictures of OSE organoids transplanted into the MFP (left panel, 5 months post-injection of 10[5] OSE organoid cells) or the ovarian bursa (right panel, 4 months post-injection of 10[5] OSE organoid cells) of *nu/nu* mice. **i** % of tumors formed in mice within 6 months after injection of 10[5] cells from wild type, LP, LP, or LPT organoids (all 4-OHT-treated), n = 10 mice/group; ***P < 0.001, Fischer's exact test. **j** and **k** Representative H&E staining and IHC for the indicated markers in LPT organoids (top) and the orthotopic tumors derived from these organoids, 4-months post injection of 10[5] cells (bottom).

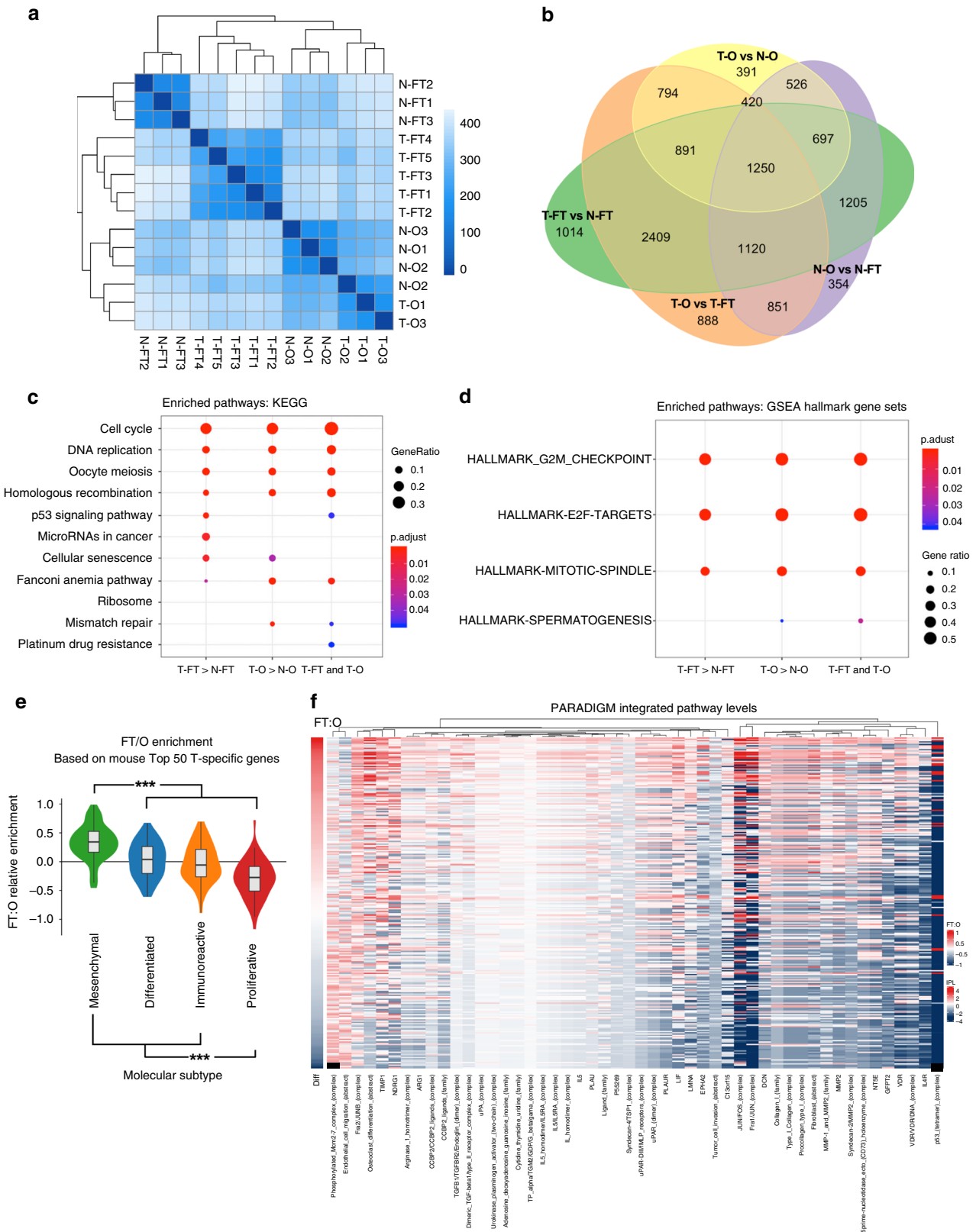

and T-O vs. T-FT, orange ovals, in Fig. 7b; also see Supplementary Fig. 8a). These included ~15% of the top 500 and 1000 DEGs, respectively (Supplementary Fig. 8b). Known FT-specific genes[53], including *Pax8* (log$_2$FC = −3.97, $P_{adj}$ = 7.68 × 10$^{-7}$; log$_2$FC =

−6.85, $P_{adj}$ < 10$^{-15}$), *Ltf* (log$_2$FC = −5.43; $P_{adj}$ = 3.24 × 10$^{-6}$; log$_2$FC = −8.72, $P_{adj}$ < 10$^{-15}$), *Slc34a2* (log$_2$FC = −4.4, $P_{adj}$ = 1.8 × 10$^{-4}$; log$_2$FC = −4.84, $P_{adj}$ = 2.7 × 10$^{-6}$), were highly expressed in N-FT and T-FT, compared with N-O and T-O. By

**Fig. 7** Comparative transcriptome analysis of mouse and human HGSOC. **a** Heatmap of sample distances by hierarchical clustering, based on total gene expression levels in normal FTE (N-FT), normal OSE (N-O), and tumors derived from FTE (T-FT), and OSE (T-O) organoids, respectively. Shading represents Euclidian distance for each sample pair. **b** Venn diagram showing number of differentially expressed genes ($P_{adj} < 0.05$) in T-FT vs. N-FT, N-O vs. N-FT, T-O vs. N-O, and T-O vs. T-FT samples. Source data are provided as a Source Data file. **c, d** Significantly enriched KEGG (**c**) and GSEA (**d**) pathways of the top 250 differentially expressed genes (DEGs) between the indicated groups. T-FT > N-FT means genes enriched in FT-derived tumors, compared with normal FT; T-O > N-O denotes genes enriched in OSE-derived tumors compared with normal OSE; T-FT and T-O means genes enriched in both tumors compared with both noro. **e** Application of cell-of-origin score, based on the top 50 DEGs in T-FT vs. T-O samples, to TCGA samples. Color coding indicates transcriptional subtype assigned to each sample by TCGA; for details, see the "Methods" section. FTE character was highest in the mesenchymal subgroup; OSE signature was highest in the proliferative subgroup, ∗∗∗$p < 0.001$, Wilcoxon rank sum test. Source data are provided as a Source Data file. **f** Top 50 PARADIGM IPLs that distinguish TCGA samples with greater T-FT character from those with more T-O character (calculated by Pearson correlation of IPLs and cell-of-origin score based on top 50 DEGs).

contrast, the known OSE-specific gene *Lgr5* ($\log_2$FC = 5.44, $P_{adj} < 10^{-15}$; $\log_2$FC = 4.41, $P_{adj} = 3 \times 10^{-15}$) had significantly higher expression in N-O and T-O than N-FT and T-FT, as did *Nr5a1* (encoding a transcriptional activator involved in sex determination/differentiation of steroidogenic tissues, $\log_2$FC = 6.52, $P_{adj} < 1 \times 10^{-15}$; $\log_2$FC = 9.62, $P_{adj} < 1 \times 10^{-15}$), *Gata4* (encoding a zinc finger transcription factor, $\log_2$FC = 5.9, $P_{adj} = 1.79 \times 10^{-6}$; $\log_2$FC = 6.4, $P_{adj} = 5.42 \times 10^{-9}$), *Lhx9* (encoding a GATA4 target, $\log_2$FC = 7.08, $P_{adj} < 1 \times 10^{-15}$; $\log_2$FC = 10.01, $P_{adj} < 1 \times 10^{-15}$), and *Unc45b* (encoding a GATA4 chaperone, $\log_2$FC = 6.23, $P_{adj} = 9 \times 10^{-11}$; $\log_2$FC = 5.72, $P_{adj} = 2.92 \times 10^{-11}$). Hence, the cell-of-origin makes a significant contribution to the tumor transcriptome.

FTE-derived and OSE-derived tumors also shared multiple DEGs, compared with their respective cells-of-origin. OSE-derived tumors had nearly 6000 DEGs, compared with normal OSE, whereas >9600 DEGs were observed in T-FT vs. N-FT (all at $P_{adj} < 0.05$) (entire green oval, including overlaps with other ovals). T-FT and T-O shared 3258 of these DEGs (Fig. 7b, Supplementary Fig. 8a, overlap between yellow and green ovals). By KEGG analysis, tumors were enriched for genes involved in the cell cycle, DNA replication, and DNA repair. Notably, p53-signaling pathway was enhanced in FTE-derived tumors, whereas several DNA repair pathways were more enriched in OSE-derived tumors (Fig. 7c). Likewise, the most enriched GO categories involved cell/nuclear division, cell cycle, and DNA replication and repair (Supplementary Fig. 9a). GSEA revealed a predominance of G2/M checkpoint control and E2F target genes in all tumor samples (Fig. 7d, Supplementary Fig. 9b); the latter comports with dysregulation of RB family/E2F regulation.

To assess the relevance of these findings to human HGSOC, we identified protein-coding genes with human orthologs (17,465 genes) in each group (T-FT, T-O, N-FT, N-O), and used differentially expressed genes (in each group, compared with all other groups) to generate group-specific signatures (see the "Methods" section and Source Data in Source Data File). Comparison of the mean scaled expression of the top (most differentially expressed) 50, 100, 250, or 500 T-FT-specific, and T-O-specific genes, respectively, with the other four groups confirmed the expected segregation of samples (Supplementary Fig. 10a–d). We developed a cell-of-origin score (see the "Methods" section), and applied it to TCGA ovarian cancer data. Whether the score was based on the top 50, 100, 250, or 500 DEGs, human tumors classified as mesenchymal were more similar transcriptionally to mouse FTE-derived tumors than to mouse OSE-derived tumors. By contrast, proliferative-type human tumors more closely resembled mouse OSE-derived tumors (Fig. 7e, Supplementary Fig. 10e). Tumors adjudged (by cell-of-origin score) more FTE- versus more OSE-like also showed differences in PARADIGM integrated pathway level (IPL) scores[54]. For example, greater FTE character was associated with higher enrichment for TP53 tetramer and FOS/JUN complex features (Fig. 7f).

Finally, we compared the somatic single nucleotide variant (SNV) and copy number abnormality (CNA) landscape in TCGA samples rated (by cell-of-origin score) more likely to derive from FTE versus OSE, respectively. Examples of all major recurrent CNAs and SNVs could be found in individual tumors inferred to originate from both cell types. Interestingly, however, there might be relative enrichment for specific CNAs in OSE-derived, and for specific SNVs in FTE-derived, tumors (Supplementary Figs. 11 and 12).

**The cell-of-origin can affect response to chemotherapeutic agents.** To ask if cell-of-origin can affect therapeutic response, we assessed the response of tumorigenic organoids to clinically used ovarian cancer drugs. PTPT (FTE) and LPT (OSE) organoids at day 4 of culture (>3 passages post-Dox or 4-OHT treatment) were released from Matrigel, ~150 organoids were re-seeded in each well of 96-well plates pre-coated with Matrigel with/without drug added, and morphology and viability were assessed 5 days later. Tumorigenic FTE and OSE organoids responded similarly to niriparib, olaparib and gemcitabine (Fig. 8a–c). By contrast, FTE-derived organoids were more sensitive to carboplatin ($P < 0.05$ at 10 μM, Sidak's multiple comparison test) and, even more prominently, to paclitaxel ($P < 0.001$ at 5, 10, and 25 μM, Sidak's multiple comparison test) treatment (Fig. 8d and e). Therefore, the cell-of-origin might influence response to the current first-line, standard-of-care therapy for HGSOC.

## Discussion

HGSOC typically presents as widely metastatic, bulky, multifocal disease, complicating identification of its cell-of-origin[55]. Previous mouse models showed that FTE can be the cell-of-origin of HGSOC, but a similar role for OSE had neither been excluded, nor demonstrated convincingly. By introducing the same genetic abnormalities into FTE or OSE in GEMMs and organoids, we establish that HGSOC can originate from either cell type (Supplementary Fig. 13). FTE-derived and OSE-derived tumors differ in latency, metastatic behavior, transcriptome, chemosensitivity, and possibly causative genomic abnormalities, suggesting that the cell-of-origin makes a substantial contribution to inter-tumor heterogeneity, molecular pathogenesis, biology, and drug response.

Similar to a previous report, in which combined *Tp53;Brca1/2; Pten* deletion was driven by *Pax8rtTA;TetOCre*[35], PTPT mice rapidly developed STIC-like lesions and metastasized early (<2 months) to the ovarian surface. Lineage tracing confirmed selective expression of *Pax8rtTA* in FTE secretory cells, but importantly, consistent with a recent study[41], we find that *Pax8*-ciliated cells derive from *Pax8*+ progenitors (Supplementary Fig. 2). Thus, while HGSOC can arise from mouse FTE, we cannot conclude that FTE *secretory* cells (as opposed to *Pax8*-derived, but *Pax8*−, ciliated cells or a specialized *Pax8*+ progenitor) are the actual/unique cell-of-origin in FTE. The *Ovgp1*

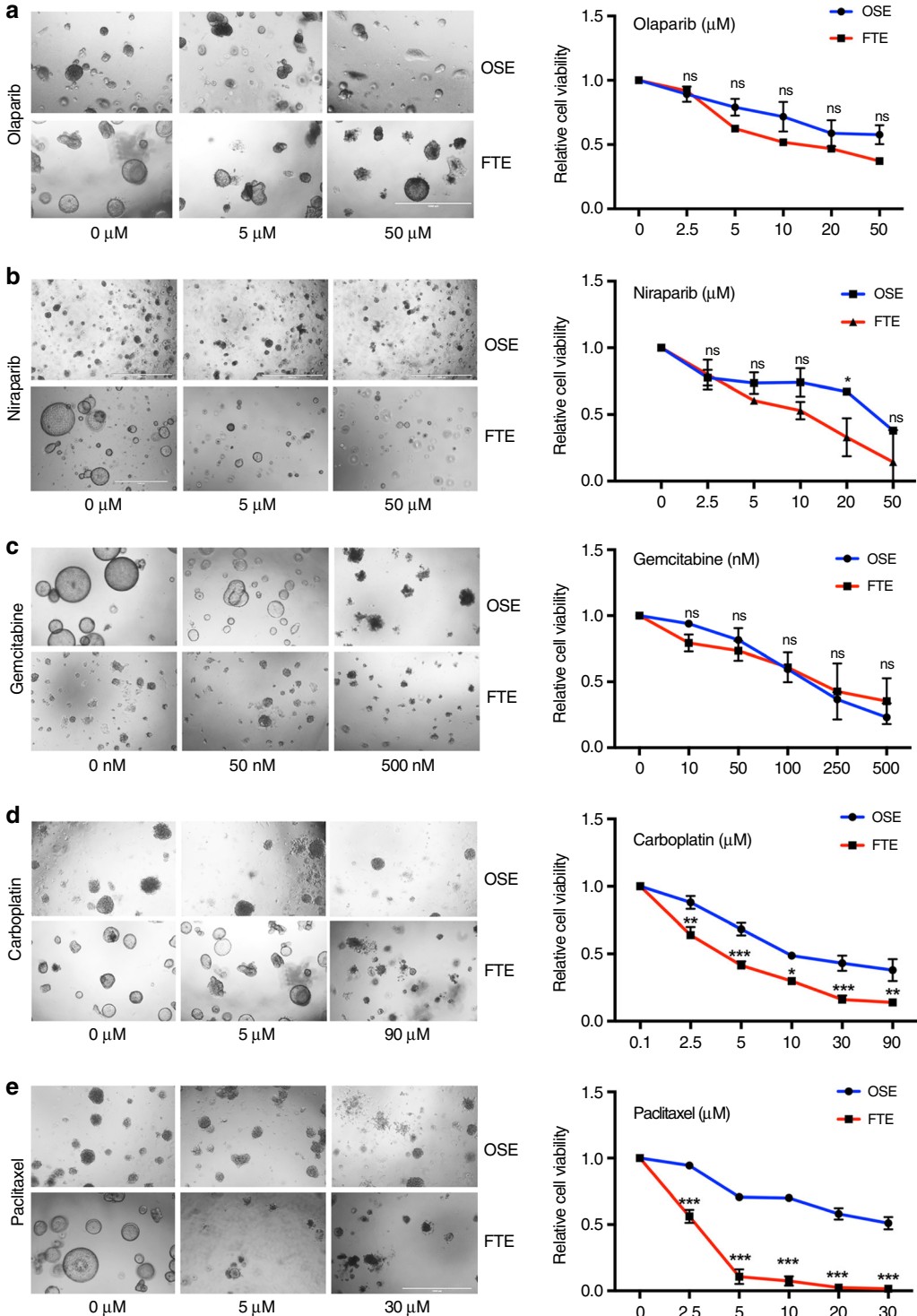

**Fig. 8** Differential response of FTE-derived and OSE-derived tumor organoids to chemotherapy. Representative micrographs and dose-response curves for LPT (OSE) and PTPT (FTE) organoids, treated with: **a** Olaparib (0–50 μM), **b** Niraparib (0–50 μM), **c** Gemcitabine (0–500 nM), **d** Carboplatin (0–90 μM), or **e** Paclitaxel (0–30 μM). Cell viability was calculated relative to 0.01% DMSO-treated control cells, measured after 5 days of treatment. Each time point represents mean ± SEM of three independent biological replicates, each in triplicate (Source data are provided as a Source Data file). *$P < 0.05$, ***$P < 0.001$, Sidak's multiple comparison test.

promoter also can drive HGSOC[36,37]. This promoter purportedly is secretory cell-specific[56], although lineage-tracing studies have not been reported. Regardless, it is now indisputable that a substantial percentage of HGSOC originates from FTE.

Whether OSE gives rise to HGSOC has been less clear. Studies in which oncogenic defects were induced by Ad-Cre injection into the ovarian bursa were interpreted as demonstrating an OSE cell-of-origin;[57,58] others argued that such injections infect adjacent tissue[34]. Such concerns were warranted: our lineage-tracing experiments show clearly that bursal injections target FTE in addition to OSE. Nevertheless, several lines of evidence demonstrate conclusively that mouse OSE can be a cell-of-origin for

HGSOC, at least HGSOC induced by combined *Tp53* mutation/ inactivation and RB family inactivation. Ad-Cre-injected *Tp53^R172H/fl*;*T121* mutant mice develop frank HGSOC even after salpingectomy. Also, *Lgr5-Cre*, which we confirm by lineage-tracing marks OSE, but not FTE, in adult mice, drives HGSOC caused by combined RB family inactivation/*Tp53* mutation/ hemizygosity. OSE organoids with the same genetic defects also give rise to HGSOC.

*Brca1/2;Pten;Tp53* deletion driven by the same *Pax8rtTA; TetOCre* transgene that we used causes STIC, invasive HGSOC, and ultimately, peritoneal carcinomatosis[35]. By contrast, PTPT mice died prematurely from thymic hyperplasia, and subsequent analysis revealed unexpected, leaky expression of *TetOCre* in thymic epithelial cells. Previous work showed that active E2F, resulting from RB family inactivation, increases *Foxn1* expression, which drives thymic epithelial cell proliferation[42]. Presumably, *Brca1/2;Pten;Tp53* deletion does not cause such hyperproliferation, explaining the absence of thymic hyperplasia in these mice[35]. However, these genes might affect other thymic epithelial cell properties and, indirectly, immune function, so our results argue for caution in using the *TetOCre* line to study tumor biology and oncogenesis.

Although we could not evaluate metastasis in PTPT mice, organoid transplantation studies show that mouse FTE-derived HGSOC metastasizes widely. We also used organoids to directly compare tumorigenesis by FTE-derived and OSE-derived HGSOC. These experiments strongly suggest that the cell-of-origin can influence HGSOC behavior: FTE-derived tumors had greater propensity to disseminate, whereas OSE-derived HGSOC formed large, solitary lesions that showed less frequent (although nonetheless significant) metastasis. Human HGSOC also differs in its pattern of growth and metastasis, with some cases forming large primary tumors and fewer, less diffuse tumor deposits, whereas others show peritoneal carcinomatosis[59]. It will be interesting to see if such differences reflect distinct cells-of-origins, which can now be assessed by transcriptomic[21,22,24], proteomic[23], and possibly epigenomic[22] analysis of such tumors.

Although we only studied the effects of RB family inactivation/ *Tp53* mutation in FTE and OSE, other defined combinations of genetic defects (e.g., as found in TCGA) can be engineered into organoids, allowing the transformation process (proliferation, polarity, cell death, invasion) to be deconstructed in vitro and tumorigenicity to be assessed in vivo. It will be important to determine whether the different tumor properties that we observe reflect the particular combination of cell-of-origin/oncogenic defects vs. the cell-of-origin per se and if all mutational combinations are equally transforming in both cell types. Notably, Hao et al. concluded that some genetic defects are more frequent in tumors likely derived from FTE vs. OSE[24], although importantly, consistent with our results, virtually all genetic abnormalities associated with human HGSOC were found in tumors inferred to arise from either cell type. By introducing these mutations into organoids, we can directly assess their relative transforming potency and biological effects in future studies. Nevertheless, RNAseq suggests that while RB inactivation/*Tp53* mutation affect similar pathways and processes in FTE and OSE, the cell-of-origin influences the specific genes affected. These transcriptional differences might contribute to differential sensitivity to carboplatin and paclitaxel: for example, while repair genes are differentially expressed in FTE-derived and OSE-derived tumors (compared with their cognate normal tissues), expression is altered more in the latter. By contrast, signaling from p53-dependent pathways (presumably by alternative mechanisms, given the combined *Tp53* mutation/deletion in both types of tumors) is elevated to a greater extent in FTE-derived, than in OSE-derived tumors.

Several lines of evidence suggest that our results are relevant for human HGSOC. A recent case report documented the development of stage IV HGSOC 3 years post-salpingectomy without evident STIC in the excised FT[60]. Recent exome sequencing[25], proteomic[23], transcriptomic[21,24], and ChIPseq experiments[22] support the notion that HGSOC can arise at sites other than FTE. These studies also concluded that HGSOC inferred to arise from OSE has a worse prognosis than putatively FTE-derived tumors. Our results provide unambiguous, biological validation of the predictions of these studies. The primary determinant of survival in HGSOC patients is their response to platinum/taxol-based chemotherapy[23,24]. Hence, the lower chemo-responsiveness of OSE-derived HGSOC organoids could explain the poorer prognosis of OSE-like HGSOC, while future studies using organoid models could be used to unravel the molecular basis of this differential response. Our findings clearly demonstrate the need to understand how the cell-of-origin contributes to HGSOC pathogenesis, given that OSE-derived and FTE-derived tumorigenic organoids have distinct biologic behavior, including sensitivities to front-line anti-neoplastic drugs. Finally, our analyses, along with those of Hao et al.[24], raise the possibility that OSE and FTE might be differentially sensitive to transformation by specific genetic combinations and different mutational processes.

## Methods

**Mouse strains**. *Rosa26-tdTomato* [B6;129S6-Gt(ROSA)26SOR^tm9(CAG-tdTomato)Hze], [B6;129-Gt(ROSA)26SorT^M1sor], *Lgr5–EGFP–ires–CreER* (Lgr5-Cre) [B6.129P2-Lgr5^tm1(cre/ERT2)Cle], *Tp53^R172H* [B6.129S4(Cg)-Trp53 ^tm2.1Tyj], *Trp53^flox/flox* [FVB;129-Trp53^tm1Brn] and *nu/nu* [NU/J] mice were obtained from The Jackson Laboratory. Conditional *TgK18GT_{121}^{tg/+}* BAC transgenic mice (T121 mice) were generated earlier[33]. Briefly, T121 (N-terminal 121 amino acids of SV40 large T antigen) was directed to OSE using transgenic *TgK18GT_{121}^{tg/+}* mice, which carry a BAC containing the mouse CK 18 gene and an inserted loxP-GFP-stop-loxP (LSL) T121 cassette. This cassette ensures that T121 is only expressed after Cre exposure. *Pax8rtTA* mice (which express the reverse tetracycline-controlled transactivator (rtTA) under control of the *Pax8* promoter) and *TetOcre* (mice expressing Cre recombinase in a tetracycline-dependent manner) strains (C57/Bl6 background) also were used[35]. *Tp53^R172H*, *Trp53^flox/flox*, and *T121* mice were interbred with *Pax8rtTA* and *TetOcre* mice to obtain *Pax8rtTA;TetOcre;Tp53^R172H/fl* (PTP), *Pax8rtTA;TetOcre;T121* (PTT) and *Pax8rtTA; TetOcre; Tp53 ^R172H/fl;T121* (PTPT) mice. *Trp53^R172H*, *Trp53^flox/flox*, *T121*, and *Lgr5Cre^ERT2* were interbred to obtain *Lgr5Cre;Tp53 ^R172H/fl* (LP), *Lgr5Cre;T121*(LT), and *Lgr5Cre; Trp53^R172H/fl;T121* (LPT) mice. *Rosa26-lacz* and *Rosa26-tdTomato* mice were bred to *Pax8rtTA* and *TetOcre* strains to obtain *Pax8rtTA;TetOcre;Rosa26-LacZ* and *Pax8rtTA;TetOcre; Rosa26-tdTomato* mice. Genotyping primers are provided in the supplementary information. When indicated, mice were euthanized by $CO_2$ inhalation and FT and/ or ovaries were harvested for histology and organoid culture. All animal experiments were approved by, and conducted in accordance with the procedures of, the IACUC at New York University School of Medicine (Protocol no.170602).

**Animal experiments**. For lineage tracing, *Pax8rtTA;TetOcre;Rosa26-LacZ* females (6–8 weeks old) were given 2 mg/ml Dox  in their drinking water for 2 days. Two (2) or 60 days later, mice were sacrificed, and their reproductive systems were collected. Before administering 4-OHT, mice were superovulated by injection of 5 IU pregnant mare serum gonadotropin (PMSG, Sigma) and 5 IU of human chorionic gonadotropin (hCG, Sigma), spaced 48 h apart. *Lgr5-Cre; Rosa26-tdTomato* mice (6–8 weeks old) were injected intraperitoneally with 2 mg of 4-OHT in sesame oil (10 mg/ml) and sacrificed 48 h or 4 months later. For *Cre* induction in PTP, PTT, and PTPT mice, Dox (2 mg/ml) was added for 2 weeks to the drinking water of 6-week-old females, and tissues were collected for H&E and IHC one month after the end of Dox treatment. For *Cre* induction in LP, LT, and LPT mice, 6-week-old females were injected with 4-OHT, and mice were sacrificed at the indicated times.

Ovarian bursae were injected with recombinant adenovirus Ad5-CMV-Cre (Ad-Cre), purchased from the Vector Development Lab, Baylor College of Medicine. *Tp53^R172H/fl;T121* females (6–8 weeks) were superovulated, and ~1.5 days later, 10 μl virus ($10^{11}$–$10^{12}$ infectious particles/ml) were delivered into one surgically exposed bursa, with the contralateral ovary serving as a control. *Rosa26-LacZ* female mice were used to test *Cre* expression, and were euthanized 2 weeks post-injection. Where indicated, salpingectomies were performed 3 days after Ad-Cre injection, and mice were euthanized 3 months later. Animals were monitored routinely for signs of distress, poor body condition, and tumor burden, and were euthanized according to veterinary recommendations. For survival experiments, mice were monitored until death or upon recommended euthanasia.

Organoids were injected into 8-week-old *nu/nu* females. Mice were anesthetized by intraperitoneal injection of 0.2 ml xylazine hydrochloride, shaved, and cleaned with betadine. A midline dorsal incision was made, followed by an incision into the peritoneal cavity above the right ovarian fat pad. The ovary was externalized, and $0.5 \times 10^6$ cells/Matrigel mixture (1:1, 15–20 μl) was injected by inserting a 27 G needle into the fat pad. The contralateral ovary served as a control. Injected tissue was returned to the peritoneal cavity, the inner incision was sutured, and the outer incision was sealed with wound clips. Cells in Matrigel (1:1, ~15 μl) were injected into the MFP just inferior to the nipples of 6–8-week-old females with a 28 G needle (BD insulin syringe). Mice that developed tumors were euthanized at the indicated times, when tumors ulcerated or reached a maximum diameter of 20 mm, or when mice showed any signs of discomfort.

**Organoid cultures and assays.** For FTE organoids, fimbriae from wild type, PTP, PTT, and PTPT mice were dissected under a microscope, minced, and digested with Collagenase type I and 0.012% (w/v) Dispase (STEMCELL Technologies) at 37 °C for 1 h, followed by incubation in TrypLE™ Express Enzyme (Thermo Fisher Scientific) for 10 min at 37 °C and inactivation with 1% FBS in DMEM (Gibco). Dispersed FTE cells were passed through a strainer (70 μm), mixed with Matrigel (BD Bioscience), seeded, and maintained in culture[61]. After the Matrigel solidified (10 min at 37 °C incubator), culture medium was added. The medium was based on Ad+++ (AdDMEM/F12, Invitrogen; HEPES, Thermo Fisher Scientific, 100× diluted; penicillin/streptomycin, Life Technologies, Glutamax, Life Technologies, 100× diluted), supplemented with B27 (Invitrogen, 50× diluted), N2 supplement (Thermo Fisher Scientific, 100× diluted), 1.25 mM N-acetylcysteine (Sigma), 50 ng/ml EGF (Thermo Fisher Scientific), 500 ng/ml RSPO1 (Peprotech) or R-spondin-1-conditioned medium (25%, v/v), WNT3a- conditioned medium (25%, v/v) and 100 ng/ml Noggin (Peprotech). For the first 3 days after thawing, media were supplemented with 10 μM Y-27632 (Sigma-Aldrich). For OSE organoids, fat and adjacent FT were removed carefully under a microscope, and the ovaries were digested with 0.25% trypsin/EDTA (Invitrogen), followed by protease inactivation with DMEM containing 1% FBS (Gibco) at 37 °C for 30 min. Supernatants, containing cells stripped from the OSE, were seeded in Matrigel, and cultured in Ad+++ medium, supplemented with B27, 1 mM N-acetylcysteine (Sigma), WNT3a-conditioned medium (50% v/v), R-spondin-1-conditioned medium (10% v/v), 100 ng/ml Noggin (Peprotech), 12.5 ng/ml EGF, 10 mM nicotinamide (Sigma), 0.5 μM A83-01 (Tocris Bioscience), 0.5 μg/ml hydrocortisone (Sigma), and 100 nM β-estradiol (Sigma). WNT3a-conditioned and R-spondin-1-conditioned media were obtained from a commercially available cell line ATCC® CRL-2647™ (ATCC) and Cultrex® R-spondin1 (Rspo1) Cells (Trevigen), respectively.

Media were changed every 2–3 days, and organoids were passaged (~10,000 cells/well) every 6–8 days. For passaging, growth medium was removed, and Matrigel was resuspended in cold Cultrex® Organoid Harvesting Solution and transferred to a 15-ml Falcon tube, which was placed on ice for 15 min. Organoids were recovered by centrifugation at 1000×g for 5 min, and resuspended in 500 μl TrypLE Express Enzyme (Gibco) for 10 min at 37 °C. Cells were seeded as indicated for each experiment. For freezing, cells were resuspended in organoid medium with 10% DMSO and 10% FBS, cooled, and stored in liquid nitrogen.

To activate *Cre* in vitro, 0.5 μl of Dox (1 mg/ml) were added to 500 μl of freshly isolated PTP, PTT, or PTPT cells, which were plated as above. Similarly, 1 μg/ml 4-OHT was used to activate Cre in LP, LT, and LPT organoids. All data were generated at least three passages after induction. Organoid size was quantified as the surface area of horizontal cross sections. If all organoids in a well could not be measured, several random, non-overlapping images were acquired from each well using an Invitrogen™ EVOS™ FL Digital Inverted Fluorescence Microscope and analyzed by using ImageJ software. Organoid perimeters for area measurements were defined manually and by automated determination using the Analyze Particle function of ImageJ, with investigator verification of the automated determinations. Organoids touching the edges of images were excluded from counting.

To compare the organoid-forming efficiency of different genotypes, 5000 cells were seeded into a 24-well plate, organoid number was counted under a light microscope after 5–7 days in culture, and the percentage of mutant organoids formed relative to those formed by wild type cells was calculated. For in vitro growth curves, organoids were incubated in TrypLE Express (Gibco) for 15 min at 37 °C, followed by an additional 5 min digestion in dispase. Isolated cells were passed through a strainer, seeded at $2 \times 10^4$ cells/well in a 24-well plate, and placed in culture medium. At the indicated times, cells were recovered as above, and viable counts were obtained by trypan blue exclusion.

Drugs were tested in organoids based on a previous protocol[62]. Briefly, organoids in culture at day 4 were released from Matrigel and diluted to 50 organoids/μl in growth medium lacking N-acetylcysteine and Y-27632. Clear bottom 96-well plates were coated with 20 μl Matrigel before the addition of 30 μl of organoid suspension. The indicated concentrations of paclitaxel (Selleck Chem), carboplatin (Sigma), olaparib (Selleck Chem), niraparib (Selleck Chem), or DMSO (control) were added in triplicate. On day 5 of treatment, media were removed, and the Matrigel drops were suspended in 40 μl CellTiter-Glo 3D (Promega) and 80 μl advanced AdDMEM/F12, and incubated for 30 min at room temperature before luminescence was measured in a FlexStation® 3 Multi-Mode Microplate Reader. Results were normalized to DMSO controls.

Invasion was assessed by using chambers with 8 μm pore size polycarbonate membrane (Transwell) inserts (Costar). Matrigel (30 μl) was added to the chambers and allowed to solidify at 37 °C for 10 min. Wild type, PTP, PTT, or PTPT cells ($2 \times 10^4$/50 μl Ad+++ medium/well) were seeded into the Matrigel-coated top chamber and allowed to attach for 12 h, followed by the addition of 500 μl of culture medium to each well. After an additional 96 h incubation, the upper surface of the membrane was scrubbed carefully several times with a cotton swab soaked in PBS to remove non-invaded cells. The lower membrane was then rinsed carefully several times with PBS, and cells that had invaded were visualized by staining with crystal violet and counted under a microscope. Invasion was calculated as the average number of cells per 10× field, determined by Image J software.

**FACS.** Ovaries from *Lgr5-Cre* (*Lgr5–EGFP–ires–Cre^{ERT2}*) females (6–8 weeks) were digested as above, and recovered OSE cells were passed through a strainer (40 μm) to obtain single-cell suspensions. OSE cells were recovered by centrifugation at 1000×g for 5 min and resuspended in PBS containing 2% FBS, 10 μM Y-27632, (STEMCELL Technologies Inc.), and DAPI (1 μg/ml). FACS was performed immediately on a MoFlo™ XDP, and GFP^{hi} and GFP^{neg} cells were seeded at 5000/well. Organoids were counted 6 days later, and organoid forming efficiency was calculated, $n = 10$ wells/group, combined from three experiments.

**Histology and immunostaining.** Tissues were fixed in 4% paraformaldehyde (PFA) in PBS at 4 °C for 4 h. Organoids were fixed in 4% PFA for 15 min and placed in Histogel (Thermo Fisher Scientific) before tissue processing and embedding. Formalin-fixed paraffin-embedded (FFPE) tissue sections (5 μm) were de-paraffinized, rehydrated, and then stained with hematoxylin and eosin (H&E) or subjected to IHC. For antigen retrieval, slides were autoclaved in 0.01 M citrate buffer (pH 6.0). Endogenous peroxidase activity was quenched in 3% $H_2O_2$ in methanol for 15 min, and sections were blocked with 0.5% BSA–PBS for 1 h. Primary antibodies were added overnight at 4 °C, then washed with PBS (3 ×, 10 min. each), incubated with HRP-labeled secondary antibodies, and washed again with PBS (3×, 10 min each). Signals were visualized by using the HRP Polymer Detection Kit and DAB peroxidase (HRP) substrate (34002, Life Technologies). Primary antibodies included: Ki67 1:200 (ab15580, Abcam), γ-H2AX 1:500 (05-636, Thermo Fisher Scientific), CK7 1:200 (ab181598, Abcam), Stathmin1 1:200 (3352 S, Cell Signaling), P16 1:200 (sc-1661, Santa Cruz), PAX8 1:200 (10336-1-AP, Proteintech), and p53 1:800 (P53-CM5P-L, Leica). Secondary antibodies were goat anti-chicken IgY-HRP 1:200 (sc-2428, Santa Cruz) and goat anti-rabbit IgG-HRP 1:200 (sc-2030, Santa Cruz), as appropriate.

Immunofluorescence was performed on frozen tissue sections (5 μm) or whole organoids released by gently dissolving the Matrigel in ice-cold PBS. Following fixation as above, cells were permeabilized in PBS containing 0.5% Triton X-100 and blocked in PBS containing 1% BSA, 3% normal goat serum, and 0.2% Triton X-100. Primary antibodies were incubated at 4 °C overnight, and sections were washed in PBS (3×, 10 min each), followed by incubation with DAPI (2 μg/ml) and Alexa 488-conjugated, 555-conjugated, or 647-conjugated anti-chicken, anti-rabbit, or anti-mouse antibodies, as indicated. After washing, samples were mounted with ProLong Gold Antifade reagent (Life Technology). Primary antibodies were: GFP (ab13970, 1:300), WT1 (ab15249, 1:200), E-cadherin (ab15148, 1:200), all from Abcam. Secondary antibodies included (all at 1:200): goat anti-mouse IgG, Alexa Fluor® 647 conjugate (A28181, Thermo Fisher Scientific), goat anti-rabbit IgG, Alexa Fluor® 555 conjugate (A27039, Thermo Fisher Scientific), and goat anti-chicken IgY H&L, Alexa Fluor® 488 conjugate (ab150169, Abcam).

**Immunoblotting.** Organoids were released from Matrigel and the cell pellets were lysed in SDS lysis buffer (50 mM Tris–HCl pH 7.5, 100 mM NaCl, 1 mM EDTA, 1% SDS, 2 mM $Na_3VO_4$), and subjected to SDS–PAGE, followed by transfer to Immobilon-FL PVDF membranes (Millipore). Membranes were blocked in 1% BSA/TBS containing 0.1% Tween20 for 30 min, and treated with primary antibodies in blocking buffer for 1 h, followed by treatment with IRDye-conjugated secondary antibodies (LI-COR). Primary antibodies were Tp53 1:1000 (P53-CM5P-L, Leica), Erk2 1:1000 (sc-1647, Santa Cruz), T1211:1000 (Anti-SV40 T Antigen (Ab-1) Mouse mAb (PAb419) (DP01)), γ-H2AX 1:1000 (05-636, Thermo Fisher Scientific). Images were obtained by using an ODYSSEY CLx quantitative IR fluorescent detection system (LI-COR). Unprocessed images of scanned immunoblots shown in Fig. 6d, Supplementary Fig. 4a are provided in a Source Data file.

**Laser capture microdissection and RNA extraction.** Tissues were embedded in FSC 22 Clear Frozen Section Compound and frozen immediately in liquid $N_2$. Blocks were sectioned at 5–8 μm, and sections were mounted on a PEN membrane frame (Leica). Slides were air-dried for 30 min at room temperature. Laser capture was performed with a Leica LMD6000 laser microdissection system. Excised pieces were harvested into 200 μl RNase-free tubes, which were carefully recovered from the microscope, centrifuged, and placed on ice. RNA was extracted by using the miRNeasy mini Kit (Qiagen), following the manufacturer's instruction.

**RNA-sequencing and data processing.** Libraries were prepared using the Illumina TruSeq Stranded Total RNA Sample Preparation Kit and sequenced on an

Illumina HiSeq 4000 using 150 bp paired-end reads by the Perlmutter Cancer Center Genome Technology Center shared resource (GTC). Sequencing results were demultiplexed and converted to FASTQ format using Illumina bcl2fastq software. The average number of read pairs/sample was 60.3M. Data were processed by the Perlmutter Cancer Center Applied Bioinformatics Laboratory shared resource (ABL). Briefly, reads were adapter- and quality-trimmed with Trimmomatic[63] and then aligned to the mouse genome (build mm10/GRCm38) using the splice-aware STAR aligner[64]. The featureCounts program[65] was utilized to generate counts for each gene, based on how many aligned reads overlap its exons. These counts were normalized and tested for differential expression, using negative binomial generalized linear models implemented by the DESeq2 R package[66]. Statistical analysis and visualization of gene sets were performed using the clusterProfiler R package[67].

**Bioinformatic analysis.** Cell type-specific gene signatures were determined by calculating differentially expressed genes pair-wise between all four groups examined (N-FT, N-O, T-FT, T-O, six comparisons). The genes were subset to protein-coding and available human ortholog predictions in any of the 14 databases indexed by the HUGO Gene Nomenclature Committee Comparison of Orthology Predictions resource. For each group, we identified genes that were upregulated relative to each of the other groups. Each gene has three P-values (indicative of the significance of upregulation of that gene in the group of interest vs. each other group). We then selected the highest (least significant) P-value, and then ranked all genes in ascending order to generate signatures of a specific size (50, 100, 250, 500 genes) for the T-FT and T-O groups, respectively. To measure enrichment in each sample, $\log_2$-transformed counts per million (logCPM) were computed for each dataset using edgeR. The logCPM matrix was scaled and centered to generate a z-score for each gene, and the mean z-score of all genes comprising each signature was calculated. Combined cell-of-origin scores (FT-O) were determined by subtracting the OSE score from the FTE score.

**Quantification and statistical analysis.** Unless otherwise specified, data are presented as mean ± SEM. Survival rates were analyzed by log-rank test, using GraphPad Prism software. P values were determined by two-tailed Student's t-test, unless otherwise specified, with $P < 0.05$ considered statistically significant.

**Reporting summary.** Further information on research design is available in the Nature Research Reporting Summary linked to this article.

## Data availability
RNA sequence data have been deposited in the GEO database under the accession code GSE125016. TCGA ovarian cancer batch effects-normalized mRNA data, somatic mutations, tumor gene-level copy number data, PARADIGM pathway activity, and molecular subtypes referenced during the study are available in a public repository from the UCSC Xena Pan-Cancer Atlas hub. The data underlying Figs. 2f, h, i, 3c, 5e, f, 6c, d, f, g, 7b, e, 8 and Supplementary Figs. 3a, b, 4a, b, c, 7a, c, 8a, 10e, 11, 12 are provided as a Source Data file. All other data supporting the findings of this study are available within the article, the Supplementary information files, or the corresponding author upon request. A reporting summary for this article is available as a Supplementary Information file.

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

## Acknowledgements

We thank Dr. Terry Van Dyke for sharing T121 mice, the PCC Experimental Pathology, Microscopy, GTC, and ABL shared resources (P30CA016087) for technical support, and Drs. Jiyuan Hu (Division of Biostatistics), Chan Wang (Division of Biostatistics), Kwan Ho Tang (PCC), and Victor Ho (Princess Margaret Cancer Center) for advice and discussion. Initial work on this project was supported by grant-MOP-191992 from the Canadian Institutes for Health Research to B.G.N., further funds were provided by Cancer Center Core Grant P30 CA016087. D.A.L. is supported by the Department of Defense Ovarian Cancer Research Program (W81XWH-15-1-0429), S.Z. was supported by a post-doctoral fellowship from Ovarian Cancer Research Fund Alliance

## Author contributions

S.Z. and B.G.N. designed the experiments, and S.Z. performed the majority of the experiments. I.D. performed the computational analyses. T.Z. performed organoid cultures and immunostaining of some samples. H.R. assisted with superovulation and salpingectomy. B.G.N., S.Z., I.D. and D.A.L wrote the manuscript.

## Competing interests

B.G.N. is a co-founder, holds equity in, and received consulting fees from Navire Pharmaceuticals and Northern Biologics, Inc. He also is a member of the Scientific Advisory Board, and receives consulting fees and equity from Avrinas, Inc, and is an expert witness for the Johnson and Johnson ovarian cancer talc litigation. His spouse has equity in Mirati Therapeutics, Amgen, Inc., Arvinas, Inc. and Array Biopharma. The other authors declare no competing interests.
