## [Peer Review File · Nature Communications]

Reviewers' comments:

Reviewer #1 (Remarks to the Author):

Zhang et al. reexamined the question whether fallopian tube (FTE) and/or ovarian surface epithelium (OSE) are the cells of origin for high-grade serous ovarian carcinoma (HGSOC) in humans. Using a series of genetically engineered mouse models and organoid systems, the authors demonstrated that ovarian carcinoma can arise from both cell types. The genetic studies are well conceived and executed. However, I feel that the manuscript is very technical and may not be an easy read for readers who are not familiar with the subject.

The authors showed that LPT mice developed multifocal peritoneal carcinomatosis in ascites resembling human HGSOC. These tumors showed little PAX8 staining, suggesting that they were of FTE origin. One naïve question - do ovarian carcinoma originated from OSE stain high for PAX8? In other words, do tumors originated from OSE ever express PAX8?

The authors carried out RNA-seq analyses on normal and tumor samples from PTPT and LPT mice. The authors carried out all possible comparisons and identified thousands of differentially expressed genes for each comparison. The authors stated that “the top 50 and top 100 differentially expressed genes in a supervised analysis of FTE-derived and OSE-derived tumors were predominantly gene differentially expressed in N-FT and N-O,..”. Since the differential analysis was done between normal and tumor samples, of course they were differentially expressed in N-FT and N-O. I think the authors meant to say that the majority of the top-ranked genes were lineage specific. In the next sentence, “N-FTE” should be “N-FT”.

The description of the gene signature analysis should be made clear – I was unable to understand what exactly the authors did. The authors stated that “FT/O-specific gene signatures were determined by calculating differentially up-regulated genes pair-wise between all four examined genes”. This sounds like that authors did all six comparisons (four groups, pair-wise). Next, the authors said it was across all three comparisons. Why choose the one with the highest p-value? Do you mean the smallest p-value? What does “scaled and centered” mean – across all samples or all genes? I assume it was across all genes in a sample, but please clarify. The authors indicated that they used “normalized counts” data. Were those HTseq data from TCGA? The authors computed OSE and FTE scores, separately. Would it make sense to identify the most significantly up-regulated genes for OSE and FTE (using T-O vs N-O and T-FT vs N-FT) first, then select the top genes as the signatures?

The authors computed the FT/O enrichment score for their own 14 RNA-seq samples using both expression data and signatures obtained from the same RNA-seq data. This is circular – gene signature obtained from RNA-seq data were applied back to the same data to compute for enrichment score. It would be helpful if author could elaborate their rationale.

Reviewer #2 (Remarks to the Author):

In this manuscript the authors assess genetic changes that have been previously targeted to the ovarian surface epithelium (OSE, by ovarian sub-bursal injection of AdCRE; Ref 33) now targeting the changes to the fallopian tube secretory epithelium (FTE; by Pax8-Cre). They provide some evidence that OSE targeting by injection of ADCRE also targets FTE, whereas Pax8 targeting of FTE excludes OSE. They show cellular transformation and early metastasis to the ovary with Pax8 restricted Cytokeratin 18 targeted S40 Tag (T121) in combination with ubiquitous R172H mutation of TP53 and Pax8 restricted loss of the wild type TP53 allele (PTPT model). However, the model succumbs to early death due to thymic complications and is thus not a good model for longitudinal or advanced ovarian cancer studies. Cre activation of the same genetic changes by the Lgr5 promoter which they confirmed was expressed in OSE, also resulted in tumor formation with abdominal masses by 11 months.

To circumvent the lethal phenotype using the Pax8 promoter and to directly compare ovarian cancer initiation in OSE and FTE, the authors establish organoid cultures with in vitro activation of Cre expression (Pax8 or Lgr5). FTE derived PTPT organoids injected into immunocompromised mice were shown to develop HGSO. OSE derived equivalent LPT organoids also induced tumor when injected into the mammary or ovarian fat pads. Organoid cultures from different origins showed different responses to some chemotherapies.

Transcriptional profiling was performed to understand the differences between OSE and FTE originating spontaneous ovarian tumors. Tumors were significantly defined by their cell of origin.

This is an important manuscript for the ovarian cancer research community at multiple levels. The question addressed in the title is critical to research. Not knowing the cell of origin of the cancer is a major confounder for establishing research models. The authors use of various genetically engineered mouse models (GEMMs) and specific CRE transgenes provides strong evidence to support the prevailing view that high grade serous ovarian cancer can arise from either the secretory cells of the fimbriae of the fallopian tube or the ovarian surface epithelial cells. Their characterization of these different GEMMs and organoid cultures and grafted immunocompromised models derived from these GEMMs fills a critical void by providing the research community with additional (organoid) models for the study of this cancer. This characterization also greatly improves our understanding of the implications of the site of origin for ovarian cancer.

Comments

Lgr5 is also expressed in other adult epithelia - eg colon (PMID 19478326). As such, the observed tumors and multifocal peritoneal carcinomatosis could be attributable to other abdominal malignancies such as colorectal cancer. How were these other possible sites of initiation eliminated in favor of an ovarian origin in the mice assessed? For example, inclusion of H&E stains of histology showing normal colon / intestinal etc, histology would be helpful.

Overall, the addition of survival curves for the different models would assist understanding of the lifespan and penetrance of different models (Pax8 spontaneous, Lgr5 spontaneous, organoid grafted). Currently graphs show % tumor formation at different times for different models and the text does not necessarily relate to the graphs. Ovarian cancer models generally suffer from incomplete penetrance and long lag times. While these factors are generally included in the text, graphical representation would be a helpful reference.

What was the stage of tumors that were used for transcriptional profiling? Same stage tumors need to be compared and Pax8-PTPT tumors are limited to STICs. Were STICs / early stage tumors compared? If not, the results would be skewed by differences in stage and the interpretation of results confounded.

For ovarian orthotopic models, organoids were engrafted into the ovarian fat pad rather than the ovarian sub-bursal space (between the ovary and the bursal membrane) where it would be exposed to the hormonal changes, secretions etc of the ovary. Please include the rationale for this site.

Regarding the Pax8rtTA;TetOCre in the OSE. Were there any Tomato + cells in the ovary / OSE? This is important to address. Suggest add image showing this.

Please add in strains / predominant background strains for the mice used.

For the results shown in Fig 1e-g and Fig 1h-j, please clarify if the animals were sacrificed 1 month after the start of Dox treatment or at different time points? It states that "as early as 1 month post-dox, metastases were detected on the OS (Fig 1h)". In the figure legend it states "... from mice with and without Dox treatment for 1 month". Both statements are ambiguous. Please ensure the timepoint that the mice were sacrificed is included in either the legend or text or figure or all. Please also ensure that the time period of Dox treatment is clearly stated as this also appears to be ambiguous.

Fig S1b: Add Rosa26-LacZ to legend for consistency

Figure 3b legend states 10^5 cells were injected. Is this correct as in methods and later in the figure the number is changed to 10^6 . (this also occurs in Figure 6). Please clarify

Fig 3b: text states 7 of 8 mice with PTPT organoids are tumorigenic but graph is different please clarify.

Fig 3c – please label define the x axis.

Fig 3f legend states that 30% of 10 mice with PTPT organoids develop metastasis, (results section states 5/7 mice within 3 months) and but graph is different again – please clarify. Similarly, Results state that 0/7 mice PTT mice developed mets, but Fig 3f states 0/10. Which is correct?

In Results - OSE-derived organoids support an ovary origin for ovarian cancer: Last line – refer to Fig 6 j and k.

Figure 4b, Top panel of each (48hour and 4 month) – indicate OSE and FTE and expand “F “and “O” in the legend.

Figure 4f – please label the x axis

What is meant by “chased” in Figure 5? Does it mean that mice were monitored for x months?

Please add in the strain of mice used in the grafted models in Figure 6.

In Methods > Animal experiments, it stated “Organoid were amplified”. Does this mean they were grown or expanded or passaged; please clarify and perhaps change wording.

In Methods > Organoid Cultures and Assays, please ensure that final concentrations are given, and this is clearly stated to prevent any ambiguity for other researchers who may want to use these conditions.

Reviewer #3 (Remarks to the Author):

The submission by Zhang et al is an extensive comparative analysis of the same genetic lesions in either Fallopian tube epithelium or ovarian surface epithelium in GEMMs and subsequent comparison of differences in tumour and non tumour phenotypes from these two putative tissues of origin of high grade serous ovarian cancer.

OSE and FTE derived tumours differed in metastatic behaviour, gene expression by transcriptome analysis and also in apparent in vitro chemosensitivity. The paper is a tour-de-force of careful GEMM analysis of the biological differences of creating p53 loss and RB loss in OSE and FTE. The biology of tumour dissemination does convincingly correlate with the biological behaviour and differences in some types of human HGSOC, although there are important differences that are not discussed in the paper. However the paper is quite hard to wade through and could really benefit from simplification overall.

The findings in my opinion add novelty to the field, with the major contribution being the proof at least in GEMM that HGSOC can arise from OSE not just from FTE, with an important observation that the biology of dissemination and features of chemosensitivity differ in OSE from FTE derived HGSOC.

The mouse studies are extensive and convincing in my opinion and stand on their own merits however workers in the field interpret the relevance of these finding to humans.

It is interesting and important that it would appear that cell type of origin would appear to determine behaviour of HGSOC to such a degree apparently.

Despite the mouse work being convincing in and of itself there are several question marks about the relevance of the work to human disease, and these should be properly and clearly addressed particularly in the introduction and discussion.

HGSOC is a collection of diseases, the biggest group by far is homologous recombination deficient HGSOC which is often regarded as a disease of Fallopian tube origin, classically described in germline BRCA mutant inherited ovarian cancer which is also exclusively p53 mutant. The model developed in this work would seem to be Rb deficient, p53 mutant disease which would correspond to a rarer group of typically non HRD HGSOC, not in the HRD group and unclear as to whether dominantly this would be principally of FTE or OSE origin. The non HRD disease assumption is underscored by the fact that sensitivity to PARP inhibitor is not different between OSE and FTE and indeed could not be considered as exquisitely sensitive to PARP inhibition in this study, and this sits at odds with the apparent sensitivity to platinum selectively in FTE. I would have expected closer correlation between PARPi and platinum sensitivity in typical HGSOC. My strong feeling is that this model therefore does not represent typical HGSOC, rather a rarer subgroup accounting for between 4 and 15% of HGSOC driven by non HRD RB deficiency. As such therefore, the broader implication may not be applicable to the majority of HGSOC and the conclusions may therefore not be secure for ovarian cancer. Whilst the transcriptomics is reassuring that there can be human ovarian cancer that looks similar to these mouse models, it is not really proof of the relationship of FTE and OSE in humans.

My strong advice to the authors to to fully describe these models in the context of HRD in the introduction and discussion. It would also help to more thoughtfully discuss the status of HRD based on transcriptomics analysis and also to consider some protein or IHC based reanalysis beyond gamma H2AX and stathmin 1 to support their premise that this model is relevant across all HGSOC.

Nevertheless I do think this paper will influence thinking in the field in that it normalises the concept that it is possible to clearly demonstrate OSE and FTE cell origins for HGSOC even if in a rare subgroup of the disease.

Then statistical analyses across the paper are valid in my opinion with the exception of the Kaplan Meier curves in supplementary figure 3 where we are told that the mice have died due to thymic enlargement by 2 months when there are clearly mice alive for much longer than this. A further

analysis could be considered for the cause of death beyond 2 months since animals are clearly alive then.

I think that the works appears reproducible generally and of high quality and have no concerns about this.

Point-by-point response to Reviewers' comments

We thank the Reviewers and the Editor for their generally positive views of our manuscript and for their perceptive comments and criticisms. In response to the Reviewers' concerns, we performed several additional experiments and include substantial new data in the revised paper. In particular, we obtained new RNAseq data from more developed tumors derived from fallopian tube and ovarian surface epithelial organoids, and compared these with their cognate normal counterparts. We also tried to better explain our lineage tracing experiments, and to clarify our bioinformatics analysis.

We believe that we have addressed all of the Reviewers' criticisms, and hope that the revised manuscript is now acceptable for publication in *Nature Communications*. Changes in the text are marked in red in the revision. In addition, Figs 3-4 and Supplementary Figs 6-12, and Supplementary Tables 1-3 are new.

Below, please find our detailed, point-by-point response to each of the Reviewers' comments (in red).

Reviewers' comments:

Reviewer #1 (Remarks to the Author):

Zhang et al. reexamined the question whether fallopian tube (FTE) and/or ovarian surface epithelium (OSE) are the cells of origin for high-grade serous ovarian carcinoma (HGSOC) in humans. Using a series of genetically engineered mouse models and organoid systems, the authors demonstrated that ovarian carcinoma can arise from both cell types. The genetic studies are well conceived and executed. However, I feel that the manuscript is very technical and may not be an easy read for readers who are not familiar with the subject.

Response: We thank the Reviewer for his/her compliments about the quality of our work. We apologize for the technical nature of the initial submission. In the revision, we have tried to make the revised manuscript as accessible as possible to the general reader.

The authors showed that LPT mice developed multifocal peritoneal carcinomatosis in ascites resembling human HGSOC. These tumors showed little PAX8 staining, suggesting that they were of FTE origin. One naïve question - do ovarian carcinoma originated from OSE stain high for PAX8? In

other words, do tumors originated from OSE ever express PAX8?

Response: PAX8 is specifically expressed in fallopian tube epithelium, not ovarian surface epithelium. Consistent with this observation, we detect little if any PAX8 staining in OSE-derived tumors (Figure 4g).

The authors carried out RNA-seq analyses on normal and tumor samples from PTPT and LPT mice. The authors carried out all possible comparisons and identified thousands of differentially expressed genes for each comparison. The authors stated that “the top 50 and top 100 differentially expressed genes in a supervised analysis of FTE-derived and OSE-derived tumors were predominantly gene differentially expressed in N-FT and N-O,..”. Since the differential analysis was done between normal and tumor samples, of course they were differentially expressed in N-FT and N-O. I think the authors meant to say that the majority of the top-ranked genes were lineage specific. In the next sentence, “N-FTE” should be “N-FT”.

Response: We thank the Reviewer for these comments. As noted by Reviewer 2, the design of the RNAseq experiments in the initial submission was problematic, because we could not be certain that we were comparing FTE- and OSE-derived tumors at similar stages of development. Therefore, we completely redid the analysis. We injected equal numbers of OSE- and FTE-derived mutant organoids into the ovarian fat pad, and performed RNAseq on tumors collected at 6 months (compared with their cognate normal tissues). These studies confirm a substantial, although lower, contribution of lineage-specific genes to differential gene expression in FTE- vs. OSE-derived tumors. Specifically, ~15% of the top 500 and 1000 DEGs in FTE-derived vs OSE-derived tumors are lineage-specific. The lower contribution of lineage genes observed in these more developed tumors could reflect tumor progression, although more likely, it reflects lower amounts of normal tissue contamination of these (larger) tumor samples. These new data are found in Fig. 7, Supplementary Figures 8-12, and Supplementary Tables 1 and 2.

The description of the gene signature analysis should be made clear – I was unable to understand what exactly the authors did. The authors stated that “FT/O-specific gene signatures were determined by calculating differentially

up-regulated genes pair-wise between all four examined genes". This sounds like that authors did all six comparisons (four groups, pair-wise). Next, the authors said it was across all three comparisons. Why choose the one with the highest p-value? Do you mean the smallest p-value? What does "scaled and centered" mean – across all samples or all genes? I assume it was across all genes in a sample, but please clarify. The authors indicated that they used "normalized counts" data. Were those HTseq data from TCGA? The authors computed OSE and FTE scores, separately. Would it make sense to identify the most significantly up-regulated genes for OSE and FTE (using T-O vs N-O and T-FT vs N-FT) first, then select the top genes as the signatures?

Response: We apologize for the lack of clarity. The Results, Methods, and Figure legends have been modified to clarify our methodology. As the Reviewer surmised, we did all six comparisons. There were two p-value-based filters. For each gene, the least significant (highest) p-value from the three pair-wise comparisons was selected. The genes were then ranked in ascending order (lowest first) of those p-values to generate gene sets of specific sizes. The approach generated signatures unique to the FTE organoid-derived tumors (T-FT) and the OSE organoid-derived tumors (T-O), relative to each other and their cognate normal tissues. The cell-of-origin signature, defined as FT-O, was used to classify TCGA tumors and infer their cell-of-origin. "Scaling and centering" was performed for each gene across all samples, as described in the revised Methods.

The authors computed the FT/O enrichment score for their own 14 RNA-seq samples using both expression data and signatures obtained from the same RNA-seq data. This is circular – gene signature obtained from RNA-seq data were applied back to the same data to compute for enrichment score. It would be helpful if author could elaborate their rationale.

Response: The Reviewer is correct. The signatures were applied to the same 14 samples, and the results were indeed as expected. However, these results are only presented (in Supplementary Fig. 10) as an internal "sanity" check on the cell-of-origin signature. This analysis generates a signature score for the individual replicates and allowed us to evaluate their variability, as well as to validate the performance of the score using an alternative normalization

method of the expression values (logCPM). The ultimate goal, however, was not to apply the signature to our own samples, but rather to evaluate the score in TCGA (and potentially other) human (or mouse) datasets, which is what we present in Fig. 7e and Supplementary 10e.

Reviewer #2 (Remarks to the Author):

In this manuscript the authors assess genetic changes that have been previously targeted to the ovarian surface epithelium (OSE, by ovarian sub-bursal injection of AdCRE; Ref 33) now targeting the changes to the fallopian tube secretory epithelium (FTE; by Pax8-Cre). They provide some evidence that OSE targeting by injection of ADCRE also targets FTE, whereas Pax8 targeting of FTE excludes OSE. They show cellular transformation and early metastasis to the ovary with Pax8 restricted Cytokeratin 18 targeted S40 Tag (T121) in combination with ubiquitous R172H mutation of TP53 and Pax8 restricted loss of the wild type TP53 allele (PTPT model). However, the model succumbs to early death due to thymic complications and is thus not a good model for longitudinal or advanced ovarian cancer studies. Cre activation of the same genetic changes by the Lgr5 promoter which they confirmed was expressed in OSE, also resulted in tumor formation with abdominal masses by 11 months. To circumvent the lethal phenotype using the Pax8 promoter and to directly compare ovarian cancer initiation in OSE and FTE, the authors establish organoid cultures with in vitro activation of Cre expression (Pax8 or Lgr5). FTE derived PTPT organoids injected into immunocompromised mice were shown to develop HGSOE. OSE derived equivalent LPT organoids also induced tumor when injected into the mammary or ovarian fat pads. Organoid cultures from different origins showed different responses to some chemotherapies. Transcriptional profiling was performed to understand the differences between OSE and FTE originating spontaneous ovarian tumors. Tumors were significantly defined by their cell of origin.

This is an important manuscript for the ovarian cancer research community at multiple levels. The question addressed in the title is critical to research. Not knowing the cell of origin of the cancer is a major confounder for establishing research models. The authors use of various genetically engineered mouse

models (GEMMs) and specific CRE transgenes provides strong evidence to support the prevailing view that high grade serous ovarian cancer can arise from either the secretory cells of the fimbriae of the fallopian tube or the ovarian surface epithelial cells. Their characterization of these different GEMMs and organoid cultures and grafted immunocompromised models derived from these GEMMs fills a critical void by providing the research community with additional (organoid) models for the study of this cancer. This characterization also greatly improves our understanding of the implications of the site of origin for ovarian cancer.

Response: We thank the Reviewer for recognizing the importance of our work.

Comments

Lgr5 is also expressed in other adult epithelia - eg colon (PMID 19478326). As such, the observed tumors and multifocal peritoneal carcinomatosis could be attributable to other abdominal malignancies such as colorectal cancer. How were these other possible sites of initiation eliminated in favor of an ovarian origin in the mice assessed? For example, inclusion of H&E stains of histology showing normal colon / intestinal etc, histology would be helpful.

Response: We thank the Reviewer for this suggestion. In the revision, we confirm that, as expected, tamoxifen treatment of *Lgr5-Cre* mice also results in expression of Tomato in intestinal epithelium (new **Supplementary Fig. 6c**). In addition, we include H&E stains for intestine and colon in the new **Supplementary Fig. 6 d**. There is no evidence of intestinal malignancy.

Overall, the addition of survival curves for the different models would assist understanding of the lifespan and penetrance of different models (Pax8 spontaneous, Lgr5 spontaneous, organoid grafted). Currently graphs show % tumor formation at different times for different models and the text does not necessarily relate to the graphs. Ovarian cancer models generally suffer from incomplete penetrance and long lag times. While these factors are generally included in the text, graphical representation would be a helpful reference.

Response: The Reviewer raises a good point, which we also considered and addressed with our organoid experiments (see **Figures 3f-g and 6h-k**). To better illustrate our results, we have included additional experiments that we had performed at the time of initial submission but did not report due to space reasons. First, equal numbers of PTPT (FTE-derived, Dox-induced) and PTL organoids (OSE-derived, 4-OHT induced) were injected into the ovarian fat pad, and survival was monitored. Equal induction of the mutant genes was assessed by flow cytometry (for GFP-). As we show in new **Supplementary Fig. 7a**, latency and survival were longer for OSE-derived tumors.

As an additional control for potentially confounding effects of Dox and/or 4-OHT, OSE- and FTE-organoids originating from $Tp53^{R172H/fl};T121$ mice were induced with Ad-Cre, and the same number of cells from each were injected into mammary fat pad to enable easy quantification of tumor size over time (scheme in new **Supplementary Fig. 8b**). As shown in the new **Supplementary Fig.7c**, OSE-derived tumors clearly grow more slowly than their FTE-derived counterparts.

What was the stage of tumors that were used for transcriptional profiling? Same stage tumors need to be compared and Pax8-PTPT tumors are limited to STICs. Were STICs / early stage tumors compared? If not, the results would be skewed by differences in stage and the interpretation of results confounded.

Response: The Reviewer raises another important point, which was inadequately addressed by our initial experiments. For this reason, as described in response to Reviewer 1, we repeated the RNAseq and bioinformatic analyses using FTE- and OSE-organoid-derived tumors arising from injection of equal numbers of cells at 6 months. These data appear in the new **Figure 7** and **Supplementary Figures 7-12**.

For ovarian orthotopic models, organoids were engrafted into the ovarian fat pad rather than the ovarian sub-bursal space between the ovary and the bursal membrane) where it would be exposed to the hormonal changes, secretions etc of the ovary. Please include the rationale for this site.

Response: We apologize for not being more clear on this point. We reported previously that primary human HGSOc cells implanted into the mouse mammary fat pad (MFP) recapitulate HGSOc histomorphology, inter- and intra-tumor heterogeneity, transcriptome, and patient drug response (*Stewart JM, PNAS, 2011; Cybulska P et al., Am J Pathol, 2018*). The MFP site facilitates accurate quantification of tumor size. Hence, as an initial test of their tumorigenic capacity, we injected wild type and mutant organoids into the MFPs of *nu/nu* mice. However, to confirm that the FTE- and OSE-derived organoids were tumorigenic in the autochthonous site, we also performed ovarian bursal injections (**Figure 3e** and **3f**, right panel in **Figure 6h**, **Figure 6i** and **Supplementary Fig. 7a**). We also clarify our reasons for using the MFP site in the revised text.

Regarding the Pax8rtTA;TetOcre in the OSE. Were there any Tomato + cells in the ovary /OSE? This is important to address. Suggest add image showing this.

Response: In **Supplementary Fig. 1**, we showed that *Pax8* is expressed specifically in FTE, not OSE, by X-gal staining of sections from *Pax8rtTA;TetOcre;LacZ* genital tracts. *Pax8rtTA;TetOcre;Tdtomato* mice were used to investigate the ontogeny of the cell types that comprise the *Pax8* lineage (**Supplementary Fig. 2**). As β -gal staining is probably more sensitive than Tomato fluorescence, we did not show the OSE in the Tomato mice in the initial paper, nor do we plan to show these data in the revision. For the Reviewer's reference, however, we present the cognate Tomato images below, which are in agreement with the X-gal staining and show FTE-specific expression of the *Pax8* transgene.

Pax8rtTA;TetOcre;Rosa26-tdTomato

Tomato DAPI

Please add in strains / predominant background strains for the mice used.

Response: All strain information is presented in the **Methods** section.

For the results shown in Fig 1e-g and Fig 1h-j, please clarify if the animals were sacrificed 1 month after the start of Dox treatment or at different time

points? It states that “as early as 1 month post-dox, metastases were detected on the OS (Fig 1h)”. In the figure legend it states “ ... from mice with and without Dox treatment for 1 month”. Both statements are ambiguous. Please ensure the timepoint that the mice were sacrificed is included in either the legend or text or figure or all. Please also ensure that the time period of Dox treatment is clearly stated as this also appears to be ambiguous.

Response: We apologize if this wording was unclear. All of the mice were analyzed at 1 month after completion of the Dox treatment. Because this was earliest time that we examined the mice, we used the phrase “as early as 1 month,” in the initial version of the paper, to indicate that metastasis might have occurred even earlier. As the Reviewer found this confusing, we have clarified the exact timing in the text and legends of the revision.

Fig S1b: Add Rosa26-LacZ to legend for consistency

Response: Done.

Figure 3b legend states 10^5 cells were injected. Is this correct as in methods and later in the figure the number is changed to 10^6 . (this also occurs in Figure 6). Please clarify

Response: We thank the Reviewer for noticing this discrepancy. All injections used 10^5 cells. We have corrected the relevant parts of the text and figures (see revised Figure 3b and Figure 6) and context.

Fig 3b: text states 7 of 8 mice with PTPT organoids are tumorigenic but graph is different please clarify.

Response: We thank the Reviewer for noticing this error as well. The correct number is 9 out of 10. We had added additional mice after our initial posting on BioRx and forgot to correct this part of the manuscript.

Fig 3c – please label define the x axis.

Response: We have labeled the X axis in the revised manuscript.

Fig 3f legend states that 30% of 10 mice with PTPT organoids develop metastasis, (results section states 5/7 mice within 3 months) and but graph is different again – please clarify. Similarly, Results state that 0/7 mice PTT mice developed mets, but Fig 3f states 0/10. Which is correct?

Response: We thank the Reviewer for pointing out these errors, which again were the result of increasing the number of mice analyzed. The correct figure is that 70% (7/10) develop metastasis, as was presented in the figure. We have corrected the relevant text.

In Results - OSE-derived organoids support an ovary origin for ovarian cancer:
Last line – refer to Fig 6 j and k.

Response: We have corrected this error in the revised manuscript.

Figure 4b, Top panel of each (48hour and 4 month) – indicate OSE and FTE and expand “F” and “O” in the legend.

Response: We are not certain what the Reviewer means here, but we have clarified the meaning of “F” (fallopian tube) and “O” (ovary) in the legend.

Figure 4f – please label the x axis

Response: We have labeled the X axis in the revised manuscript.

What is meant by “chased” in Figure 5? Does it mean that mice were monitored for x months?

Response: This is a lineage-tracing experiment, so the cells of interest are “marked” at one time (“pulse”), and the progeny derived from these marked cells are revealed at a later timepoint (“chase”). However, to avoid jargon that might confuse the general reader, we have rewritten/clarified the description of these experiments in the figure legend.

Please add in the strain of mice used in the grafted models in Figure 6.

Response: We have included this information in the revised Figure legend.

In Methods > Animal experiments, it stated "Organoid were amplified". Does this mean they were grown or expanded or passaged; please clarify and perhaps change wording.

Response: We have changed the wording to "grown" in the revision.

In Methods > Organoid Cultures and Assays, please ensure that final concentrations are given, and this is clearly stated to prevent any ambiguity for other researchers who may want to use these conditions.

Response: We have included the requested information in the revised version of Methods.

Reviewer #3 (Remarks to the Author):

The submission by Zhang et al is an extensive comparative analysis of the same genetic lesions in either Fallopian tube epithelium or ovarian surface epithelium in GEMMs and subsequent comparison of differences in tumour and non tumour phenotypes from these two putative tissues of origin of high grade serous ovarian cancer.

OSE and FTE derived tumours differed in metastatic behaviour, gene expression by transcriptome analysis and also in apparent in vitro chemosensitivity. The paper is a tour-de-force of careful GEMM analysis of the biological differences of creating p53 loss and RB loss in OSE and FTE.

Response: We thank the Reviewer for his/her very nice comments.

The biology of tumour dissemination does convincingly correlate with the biological behaviour and differences in some types of human HGSOE, although there are important differences that are not discussed in the paper. However the paper is quite hard to wade through and could really benefit from simplification overall.

Response: Reviewer 1 had similar concerns. In the revision, we have endeavored to simplify our story to the extent possible, given the intrinsic sophistication of some of these models. Of course, if the Editor/Reviewer has specific concerns about the revision, we would be happy to make further corrections/modifications.

The findings in my opinion add novelty to the field, with the major contribution being the proof at least in GEMM that HGSOC can arise from OSE not just from FTE, with an important observation that the biology of dissemination and features of chemosensitivity differ in OSE from FTE derived HGSOC. The mouse studies are extensive and convincing in my opinion and stand on their own merits however workers in the field interpret the relevance of these finding to humans. It is interesting and important that it would appear that cell type of origin would appear to determine behaviour of HGSOC to such a degree apparently.

Response: We again thank the Reviewer for his/her complimentary comments.

Despite the mouse work being convincing in and of itself there are several question marks about the relevance of the work to human disease, and these should be properly and clearly addressed particularly in the introduction and discussion.

HGSOC is a collection of diseases, the biggest group by far is homologous recombination deficient HGSOC which is often regarded as a disease of Fallopian tube origin, classically described in germline BRCA mutant inherited ovarian cancer which is also exclusively p53 mutant. The model developed in this work would seem to be Rb deficient, p53 mutant disease which would correspond to a rarer group of typically non HRD HGSOC, not in the HRD group and unclear as to whether dominantly this would be principally of FTE or OSE origin. The non HRD disease assumption is underscored by the fact that sensitivity to PARP inhibitor is not different between OSE and FTE and indeed could not be considered as exquisitely sensitive to PARP inhibition in this study, and this sits at odds with the apparent sensitivity to platinum selectively in FTE. I would have expected closer correlation between PARPi and platinum sensitivity in typical HGSOC. My strong feeling is that this model

therefore does not represent typical HGSOC, rather a rarer subgroup accounting for between 4 and 15% of HGSOC driven by non HRD RB deficiency. As such therefore, the broader implication may not be applicable to the majority of HGSOC and the conclusions may therefore not be secure for ovarian cancer. Whilst the transcriptomics is reassuring that there can be human ovarian cancer that looks similar to these mouse models, it is not really proof of the relationship of FTE and OSE in humans.

My strong advice to the authors to to fully describe these models in the context of **HRD in the introduction and discussion**. It would also help to more thoughtfully discuss the status of HRD based on transcriptomics analysis and also to consider some protein or IHC based reanalysis beyond gamma H2AX and stathmin 1 to support their premise that this model is relevant across all HGSOC.

Nevertheless I do think this paper will influence thinking in the field in that it normalises the concept that it is possible to clearly demonstrate OSE and FTE cell origins for HGSOC even if in a rare subgroup of the disease.

Response: We take the Reviewer's point, and for space reasons, have modified the Introduction and Discussion within the space constraints allotted. Specifically, in the revised Discussion, we note that we cannot ensure that all genotypes would be equally able to generate HGSOC from FT and OSE. That said, we also note that a previous study used gene signatures to identify human tumors more likely to have initiated from FTE or OSE, respectively, and found few significant differences in the genetic defects in such tumors (Hao *et al.* Clin. Ca Res. **30**, 7400-7411, 2017). Likewise, in our revised manuscript, we applied find examples of all major HGSOC-associated CNVs and mutations in tumors that we classify as more FTE-like or more OSE-like by using our cell-of origin score (new **Supplementary Figs. 11 and 12**). Interestingly, this analysis also raises the intriguing possibility that OSE and FTE might be differentially susceptible to transformation by specific oncogenic insults, and might also be differentially pre-disposed to different mutational processes (i.e., CNAs might be enriched in OSE-derived tumors, whereas somatic SNVs are more frequent in FTE-derived tumors). We plan to investigate these possibilities in the future and thank the Reviewer for his/her perceptive comments, which motivated these analyses.

Then statistical analyses across the paper are valid in my opinion with the exception of the Kaplan Meier curves in supplementary figure 3 where we are told that the mice have died due to thymic enlargement by 2 months when there are clearly mice alive for much longer than this. A further analysis could be considered for the cause of death beyond 2 months since animals are clearly alive then.

Response: The Reviewer is correct that ~20% of the mice were alive at 2 months, but all of them had enlarged thymi, and they died within several days later. With respect, we do not think that further analysis of these animals would be productive.

REVIEWERS' COMMENTS:

Reviewer #1 (Remarks to the Author):

The authors have addressed my major concerns. I appreciate their efforts for making it accessible to a general reader.

Reviewer #2 (Remarks to the Author):

The Reviewer comments have largely been addressed. A few points remain:

Legend Fig 1h: there are no arrows in the figures

Legend Fig 1i: "assessed after 1 month of Dox treatment" – poorly worded, implying that Dox was given for 1 month; please correct.

Page 4, line 116: Supp Fig 2a should be 2a-2b; and line 118, Sup Fig 2b should be 2c-2d.

Legend Fig 5g: p53 IHC is not shown in the figure.

Results \ page 7 \ Similarities and differences between transcriptomes

This section remains hard to follow.

Suggest add bold lines to Fig 7b to show the areas representing 3,641 DEGS (line 261), ~9,600 DEGS (line 276) and 3,258 DEGS (line 277).

Paragraph starting at line 256 and associated Methods: what is the base of logFC? This is not stated anywhere. Is it log base2 or log base10 or?

This section should be rewritten to more easily follow what comparison each logFC refers to.

Figure 7c: the figure has 2 columns of T-O>N-O. Is this correct?

In Figures 7c,7d, Sup Fig 9, some column comparisons are written as ">" and others as "and". Please clarify as this is difficult to interpret.

Reviewer #3 (Remarks to the Author):

I am satisfied with the author responses

Point-by-point response to reviewer's comments

REVIEWERS' COMMENTS:

Reviewer #1 (Remarks to the Author):

The authors have addressed my major concerns. I appreciate their efforts for making it accessible to a general reader.

Response: We appreciate the Reviewer's comment.

Reviewer #2 (Remarks to the Author):

The Reviewer comments have largely been addressed.

Thank you for your careful attention to our manuscript. We genuinely appreciate your efforts.

A few points remain:

Legend Fig 1h: there are no arrows in the figures

Response: Thanks for pointing out this mistake, which was carried over from an earlier version. We have changed the legend accordingly.

Legend Fig 1i: "assessed after 1 month of Dox treatment" – poorly worded, implying that Dox was given for 1 month; please correct.

Response: We have changed the wording to "assessed as in h".

Page 4, line 116: Supp Fig 2a should be 2a-2b; and line 118, Sup Fig 2b should be 2c-2d.

Response: Thanks for the careful reading. We have made these corrections in the revised manuscript.

Legend Fig 5g: p53 IHC is not shown in the figure.

Response: We thank the Reviewer for noticing this error, which again was from an earlier version. We have corrected the legend.

Results \ page 7 \ Similarities and differences between transcriptomes

This section remains hard to follow.

Suggest add bold lines to Fig 7b to show the areas representing 3,641 DEGS (line 261), ~9,600 DEGS (line 276) and 3,258 DEGS (line 277).

Response: We appreciate the Reviewer's thoughtful concern. However, we think that altering the Venn diagram would be even more confusing. We have tried to provide greater clarity by including more detail about the relevant ovals being compared in the revised manuscript. We also include the UPset diagram (Supplementary Fig. 8a) in an attempt to make this information easier to follow.

Paragraph starting at line 256 and associated Methods: what is the base of logFC? This is not stated anywhere. Is it log base2 or log base10 or?

This section should be rewritten to more easily follow what comparison each logFC refers to.

Response: The log is base 2. We have reworded the relevant sections to \log_2FC .

Figure 7c: the figure has 2 columns of T-O>N-O. Is this correct?

In Figures 7c,7d, Sup Fig 9, some column comparisons are written as ">" and others as "and". Please clarify as this is difficult to interpret.

Response: Thanks for noticing this. An extra column (column 2) was left in Figure 7c from the original submission, which showed genes enriched in normal ovary vs. ovarian tumor (i.e., the converse of the third column, which shows genes enriched in ovarian tumor compared with normal ovary). The original submission had a similar comparison of genes enriched in normal FT vs FT-derived tumor (i.e., the converse of column 1). We removed the latter, but not the former in the revision. We have removed the second column, so that Fig. 7c now parallels Fig. 7d. We apologize for the error.

As far as the nomenclature is concerned, T-FT>N-FT, means genes enriched in FT-derived tumor compared with normal FT. T-O>N-O means genes enriched in OSE-derived tumor compared with normal OSE. T-FT and T-O means genes enriched in both tumors, compared with both normal. We have explained this more clearly in the text and figure legends.

Reviewer #3 (Remarks to the Author):

I am satisfied with the author responses

Response: We thank the Reviewer for his/her help in improving our manuscript.